# APEX: One-Step High-Resolution Text-to-Image Synthesis

## ABSTRACT

The pursuit of efficient text-to-image synthesis has driven the field toward a few-step generation paradigm, yet this endeavor is hampered by a persistent trilemma: achieving high fidelity, inference efficiency, and training efficiency simultaneously remains elusive. Current approaches are often forced into a difficult trade-off. While methods employing external discriminators can produce high-fidelity one-step generations, they suffer from significant drawbacks, including training instability, high GPU memory costs, and slow convergence. Conversely, alternative paradigms like consistency distillation, though easier to train, often struggle to achieve high quality in one-step generation. These challenges have restricted the scalability and broader application of one-step generative models. In this work, we present **APEX**, a method that resolves this trilemma. The core innovation is a *self-condition-shifting adversarial mechanism* that completely obviates the need for an external discriminator. By eliminating this discriminator bottleneck, APEX achieves exceptional training efficiency and stability. This design makes it particularly well-suited for both full-parameter and LoRA-based tuning of large-scale generative models, offering a truly end-to-end solution. Experimentally, APEX demonstrates state-of-the-art (SOTA) performance, delivering high-fidelity synthesis with just a single function evaluation **(NFE=1)**, yields a **15.33x** speedup over the original Qwen-Image 20B. Our **0.6B** model improves upon substantially larger models, such as FLUX Schnell **12B** in few-step generation. We further showcase its efficiency by achieving a GenEval score of **0.89** on the Qwen-Image (original 50 NFE is 0.87) for 1 NFE, **20B** model with LoRA tuning in just **6 hours**. APEX effectively reshapes the trade-off between training cost, inference speed, and generation quality in large text-to-image generative models.

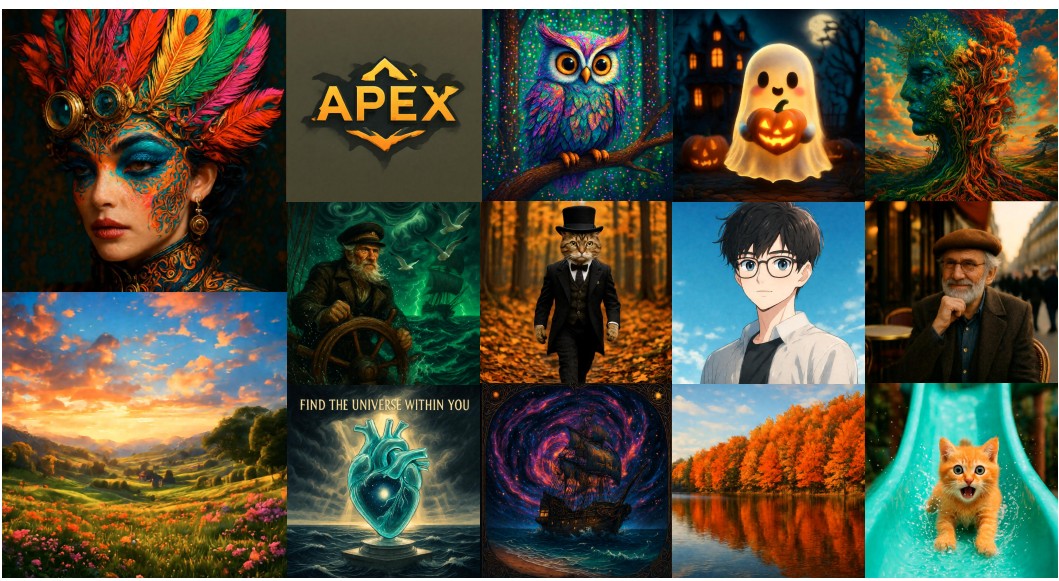

Figure 1: **An overview of generated images.**

# 1 INTRODUCTION

Continuous generative models have achieved unprecedented generative fidelity across diverse domains, from photorealistic image synthesis (Dhariwal & Nichol, 2021; Karras et al., 2024) to coherent video generation (Ho et al., 2022; Chen et al., 2025b) and interactive world modeling (Deepmind, 2025). The foundational engine driving this remarkable progress is the iterative simulation of stochastic differential equations (SDEs) or their deterministic Probability Flow Ordinary Differential Equations (PF-ODEs) (Song et al., 2020), as exemplified by diffusion models (Ho et al., 2020; Dhariwal & Nichol, 2021) and flow-matching frameworks (Lipman et al., 2022; Ma et al., 2024). These models achieve their exceptional quality by gradually transforming noise distributions into complex data manifolds through fine-grained steps. However, this iterative paradigm imposes an inherent dichotomy between generation quality and inference speed. The high fidelity achieved through multi-step simulation comes at a steep computational cost (Karras et al., 2024; Nichol & Dhariwal, 2021), severely constraining deployment in latency-sensitive applications. Consequently, the field has converged on the ambitious goal of one-step synthesis (Song et al., 2023; Salimans & Ho, 2022; Yin et al., 2024). The aim is to achieve multi-step quality within a single function evaluation (NFE=1), transforming generative models to real-time creative engines.

Realizing this ambition requires confronting a fundamental dichotomy: a tension between ensuring the numerical stability of the integration path termed **Path Integrability** and guaranteeing the perceptual quality of the final image, or **Endpoint Fidelity**. One class of methods, including diffusion and flow-matching, addresses this via *local first-order supervision*, learning the instantaneous velocity of the ODE's vector field (Ho et al., 2020; Lipman et al., 2022; Karras et al., 2024). While effective for multi-step integration, this approach remains agnostic to higher-order path geometry, particularly *path curvature*, rendering the learned trajectory brittle under coarse discretization and prone to accumulating catastrophic truncation error (Karras et al., 2022b). A second class, including consistency models, enforces a *global but perceptually weak constraint* by learning to map any point on a trajectory directly to its endpoint (Song et al., 2023; Lu & et al., 2024). This strategy is purpose-built for efficient inference, but its reliance on pixel-level regression objectives is insufficient for penalizing high-frequency artifacts, a weakness that has historically cemented adversarial training as essential for photorealism (Goodfellow et al., 2014; Sauer et al., 2023). This bifurcation of the field motivates our central goal: to resolve this trade-off within a single, unified framework.

> $\mathbb{Q}$: **How to break the trade-off between Path Integrability and Endpoint Fidelity to achieve multi-step quality at NFE=1?**

**Our approach.** In this work, we introduce **APEX**, a new method against the problem. First, to ensure **Path Integrability**, APEX imposes a *higher-order path self-consistency* constraint. Instead of merely matching instantaneous velocities, this mechanism forces the model's prediction over any large interval to be consistent with the integral of its own predicted velocities within that interval. This directly regularizes the local path curvature—the leading-order source of truncation error—promoting smoother, more integrable trajectories that remain stable even under the extreme discretization of single-step inference. Second, to guarantee **Endpoint Fidelity**, APEX introduces a discriminator-free, *self-condition-shifting* adversarial mechanism. The model learns to generate its own adversarial signal by maximizing the perceptual distance between its outputs under correct versus strategically perturbed conditioning. This internal contrast forces the generator to capture the high-frequency details and fine-grained textures most salient to the conditioning manifold, achieving GAN-level realism without the associated training instabilities or computational overhead.

**Our main contributions are:**

(a) **Theoretical Framework:** We providing a unified lens to understand existing generative paradigms and formalize the central challenge of *Path Integrability* versus *Endpoint Fidelity*.

(b) **Method:** We introduce **APEX**, a method built on two novel mechanisms: a *higher-order path self-consistency* constraint that ensures numerical stability under large step sizes, and a discriminator-free *self-condition-shifting* adversarial mechanism that enforces realism.

(c) **SOTA Performance and Efficiency:** We demonstrate unprecedented performance and efficiency in large-scale text-to-image synthesis. Experimentally, APEX delivers SOTA quality with NFE=1. Our 0.6B-parameter model improves upon substantially larger models, such as FLUX (12B). Furthermore, we showcase its scalability by achieving a GenEval score of 0.89 on the Qwen-Image (20B) model via LoRA tuning in just 6 hours, reshaping the trade-off between training cost, inference speed, and generation quality.

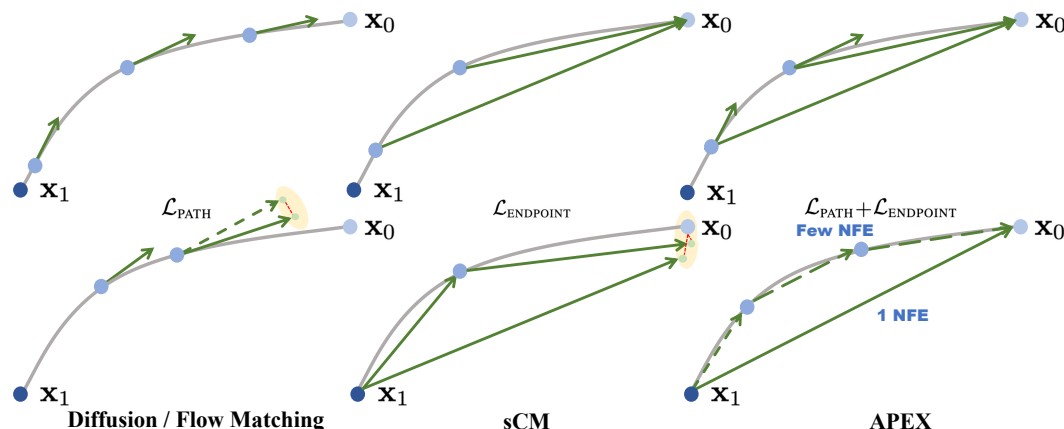

Figure 2: **Comparison between APEX and existing methods.** *Diffusion / Flow Matching* is trained via local first-order supervision to match the path's instantaneous velocity. While effective for multi-step integration, this locality makes the learned vector field brittle to large discretization steps, causing the numerical solution to accumulate truncation error and diverge from the true path. *sCM* is trained to map any point on the trajectory directly to its endpoint $\mathbf{x}_0$. This is optimized for one-step generation, but its reliance on perceptually simple objectives is insufficient to penalize high-frequency artifacts, thus compromising endpoint fidelity. *APEX* resolves this dichotomy. Its higher-order path objective ensures the trajectory remains numerically stable, while using self condition shifting adversarial guarantees the perceptually realistic endpoint.

## 2 PRELIMINARIES

### 2.1 CONTINUOUS GENERATIVE MODELS: LINEAR TRANSPORT AND PF-ODE

Continuous generative models learn a deterministic transport from a simple prior distribution $p_1(\mathbf{z}) = \mathcal{N}(0, I)$ to a complex data distribution $p_0(\mathbf{x}) = p_{\text{data}}$. This process is often conceptualized as traversing a continuous path $\{\mathbf{x}_t\}_{t \in [0,1]}$ defined by a linear Gaussian transport process (Song et al., 2020; Karras et al., 2022b):

$$\mathbf{x}_t = \alpha(t) \cdot \mathbf{z} + \gamma(t) \cdot \mathbf{x}, \qquad t \in [0, 1], \tag{1}$$

where $\alpha, \gamma : [0, 1] \to \mathbb{R}$ are continuously differentiable schedules satisfying the boundary conditions $\gamma(0) = 1$, $\alpha(0) = 0$ and $\gamma(1) = 0$, $\alpha(1) = 1$. This ensures that the path interpolates between the data distribution $p_0$ and the prior distribution $p_1$. The dynamics of this path are governed by a probability flow ordinary differential equation (PF-ODE) (Song et al., 2020), which defines the instantaneous velocity field $\mathbf{v}(\mathbf{x}_t, t)$ required to reverse the noising process.

A unified training objective (Sun et al., 2025) can be formulated where a neural network $\boldsymbol{F_\theta}(\mathbf{x}_t, t)$ is trained to regress a target signal $\mathbf{z}_t$, which is itself a linear combination of the data and noise:

$$\mathcal{L}(\boldsymbol{\theta}) := \mathbb{E}_{(\mathbf{x}, \mathbf{z}) \sim p(\mathbf{x}, \mathbf{z}), t} \left[ \frac{1}{\omega(t)} \| \boldsymbol{F_\theta}(\mathbf{x}_t, t) - \mathbf{z}_t \|_2^2 \right], \tag{2}$$

*Color convention:* we use student/current prediction and teacher/reference (EMA/stop-gradient) throughout. where $\mathbf{x}_t = \alpha(t) \cdot \mathbf{z} + \gamma(t) \cdot \mathbf{x}$ and the target is $\mathbf{z}_t = \hat{\alpha}(t) \cdot \mathbf{z} + \hat{\gamma}(t) \cdot \mathbf{x}$. The choice of coefficients $\hat{\alpha}$ and $\hat{\gamma}$ distinguishes different model families. Given a trained network, its output $\boldsymbol{F}_t := \boldsymbol{F_\theta}(\mathbf{x}_t, t)$ can be inverted to yield estimates of the clean data $\mathbf{x}$ and the initial noise $\mathbf{z}$:

$$\boldsymbol{f^x}(\boldsymbol{F}_t, \mathbf{x}_t, t) := \frac{\alpha(t) \cdot \boldsymbol{F}_t - \hat{\alpha}(t) \cdot \mathbf{x}_t}{\alpha(t) \cdot \hat{\gamma}(t) - \hat{\alpha}(t) \cdot \gamma(t)}, \quad \boldsymbol{f^z}(\boldsymbol{F}_t, \mathbf{x}_t, t) := \frac{\hat{\gamma}(t) \cdot \mathbf{x}_t - \gamma(t) \cdot \boldsymbol{F}_t}{\alpha(t) \cdot \hat{\gamma}(t) - \hat{\alpha}(t) \cdot \gamma(t)}. \tag{3}$$

At training time, one typically regresses $\mathbf{z}_t$ with $\boldsymbol{F}_t$. At inference time, Eqs. (3) above decompose $\mathbf{x}_t$ into the estimated components $\hat{\mathbf{x}}$ and $\hat{\mathbf{z}}$, which are then re-composed to form the next state via the same transport: $\mathbf{x}_{t'} = \alpha(t') \cdot \hat{\mathbf{z}} + \gamma(t') \cdot \hat{\mathbf{x}}$. This framework encapsulates major paradigms: (i) **Diffusion models** in the EDM/EDM2 family (Karras et al., 2022b; 2024) correspond to particular choices of $(\hat{\alpha}, \hat{\gamma})$ that use $\boldsymbol{F}_t$ to regress a normalized noise/residual target and then enable to sample with the PF-ODE driven by the learned $\boldsymbol{F}_t$. (ii) **Flow Matching** (Lipman et al., 2022), by contrast, sets the target to the instantaneous velocity by choosing $\hat{\alpha}(t) = \frac{\mathrm{d}\alpha(t)}{\mathrm{d}t}$ and $\hat{\gamma}(t) = \frac{\mathrm{d}\gamma(t)}{\mathrm{d}t}$, making $\mathbf{z}_t \equiv \mathbf{v}(\mathbf{x}_t, t)$ and turning the training into direct velocity fitting. During training, we randomly

sample $(\mathbf{x}, \mathbf{z}, t)$ and construct the linear interpolant $\mathbf{x}_t = \alpha(t) \cdot \mathbf{z} + \gamma(t) \cdot \mathbf{x}$ and the flow-matching target $\mathbf{z}_t = \frac{\mathrm{d}\alpha(t)}{\mathrm{d}t} \cdot \mathbf{z} + \frac{\mathrm{d}\gamma(t)}{\mathrm{d}t} \cdot \mathbf{x}$. Under the joint distribution over $(\mathbf{x}, \mathbf{z}, t)$, this linear combination matches the ideal mean velocity of a straight interpolant trajectory, yielding a directly measurable supervision signal.

While these models focus on local, first-order dynamics, **Consistency Models (CMs)** (Song et al., 2023; Lu & et al., 2024) are designed for few-step synthesis. They learn a function $\boldsymbol{f_\theta}(\mathbf{x}_t, t)$ that maps any point on a trajectory directly to its origin $\mathbf{x}_0$. This is enforced by a consistency loss between adjacent time steps, $\mathcal{L}_{\mathrm{CM}} = \mathbb{E}[d(\boldsymbol{f_\theta}(\mathbf{x}_t, t), \ \boldsymbol{f_{\theta^-}}(\mathbf{x}_{t-\Delta t}, t - \Delta t))]$, where $d(\cdot, \cdot)$ is a metric function. In the continuous-time limit ($\Delta t \to 0$), this objective simplifies to an inner product that aligns the student's prediction with the direction of the teacher's change along the ODE path (Lu & et al., 2024):

$$\mathcal{L}_{\mathrm{CM}} = \mathbb{E}_{\mathbf{x}_t, t}\left[ w(t) \left\langle \boldsymbol{f_\theta}(\mathbf{x}_t, t), \ \tfrac{\mathrm{d}}{\mathrm{d}t} \boldsymbol{f_{\theta^-}}(\mathbf{x}_t, t) \right\rangle \right]. \tag{4}$$

where the total time derivative, $\frac{\mathrm{d}}{\mathrm{d}t} \boldsymbol{f_{\theta^-}} = \partial_t \boldsymbol{f_{\theta^-}} + \nabla_{\mathbf{x}_t} \boldsymbol{f_{\theta^-}} \cdot \frac{\mathrm{d}\mathbf{x}_t}{\mathrm{d}t}$, captures both explicit dependence on $t$ and implicit dependence through the state $\mathbf{x}_t$'s movement along the path. In practice, $\frac{\mathrm{d}}{\mathrm{d}t}\boldsymbol{f}$ can be computed by a JVP operator (Lu & et al., 2024) or, for scalability, by a forward-pass-only central-difference estimator (Wang et al., 2025).

## 2.2 GANs and Adversarial Objectives

Generative Adversarial Networks (GANs) (Goodfellow et al., 2014) directly optimize the data generation process through a two-player minimax game, without explicitly modeling the data likelihood. The framework involves a generator network $G_{\boldsymbol{\theta}}$ that maps latent noise to synthetic samples, and a discriminator network $D_{\boldsymbol{\phi}}$ trained to distinguish real samples from generated ones:

$$\min_{\boldsymbol{\theta}} \max_{\boldsymbol{\phi}} \ \mathbb{E}_{\mathbf{x} \sim p_{\mathrm{data}}}\left[ \log D_{\boldsymbol{\phi}}(\mathbf{x}) \right] + \mathbb{E}_{\tilde{\mathbf{x}} \sim p_{\boldsymbol{\theta}}}\left[ \log(1 - D_{\boldsymbol{\phi}}(\tilde{\mathbf{x}})) \right]. \tag{5}$$

where $p_{\boldsymbol{\theta}} = G_{\boldsymbol{\theta}}(\mathbf{z})$ and $\mathbf{z} \sim \mathcal{N}(0, I)$. For a fixed generator, the optimal discriminator $D^*(\mathbf{x}) = \frac{p_{\mathrm{data}}(\mathbf{x})}{p_{\mathrm{data}}(\mathbf{x}) + p_{\boldsymbol{\theta}}(\mathbf{x})}$. Substituting this reveals that the generator is implicitly minimizing the Jensen-Shannon (JS) divergence between the model and data distributions. While powerful, GAN training often faces challenges like instability and mode collapse, leading to variants like Wasserstein GANs (Arjovsky et al., 2017) that improve stability. The principles of adversarial training are increasingly incorporated to enhance other generative models. For instance, a GAN loss can serve as an auxiliary objective to improve one-step diffusion distillation (Kim et al., 2023b; Yin et al., 2024).

## 3 APEX

The analysis in Section 2 establishes that learning an integrable transport path is a central task in continuous generative modeling. However, the challenge of ensuring high **Endpoint Fidelity** in few-step generation remains. To resolve this, we introduce **APEX**, a framework that explicitly optimizes for both endpoint realism and smoothly integrable path through a dual-component loss:

$$\mathcal{L}_{\mathrm{APEX}}(\boldsymbol{\theta}) = \underbrace{\mathcal{L}_{\mathrm{endpoint}}(\boldsymbol{\theta})}_{\text{Ensures } \hat{\mathbf{x}}_0 \in p_{\mathrm{data}}} + \underbrace{\mathcal{L}_{\mathrm{path}}(\boldsymbol{\theta})}_{\text{Ensures path is smoothly integrable}} \tag{6}$$

### 3.1 Path Integrability Objective

The path objective is designed to produce a numerically stable vector field that is robust to large integration steps. It builds upon a unified consistency framework that establishes a gradient equivalence between an endpoint-space objective and a velocity-space regression.

**Primal and dual losses.** The starting point is the primal loss (Sun et al., 2025), which enforces self-consistency on the predicted endpoint $f^{\mathbf{x}}$ at different points in time:

$$\mathcal{L}_{\mathrm{path}}^{\mathrm{primal}}(\boldsymbol{\theta}) = \mathbb{E}_{(\mathbf{x},\mathbf{z}) \sim p(\mathbf{x},\mathbf{z}), t}\left[ \frac{1}{\hat{\omega}(t)} \left\| f^{\mathbf{x}}(\boldsymbol{F_\theta}(\mathbf{x}_t, t), \mathbf{x}_t, t) - f^{\mathbf{x}}(\boldsymbol{F_{\theta^-}}(\mathbf{x}_{\lambda t}, \lambda t), \mathbf{x}_{\lambda t}, \lambda t) \right\|_2^2 \right], \tag{7}$$

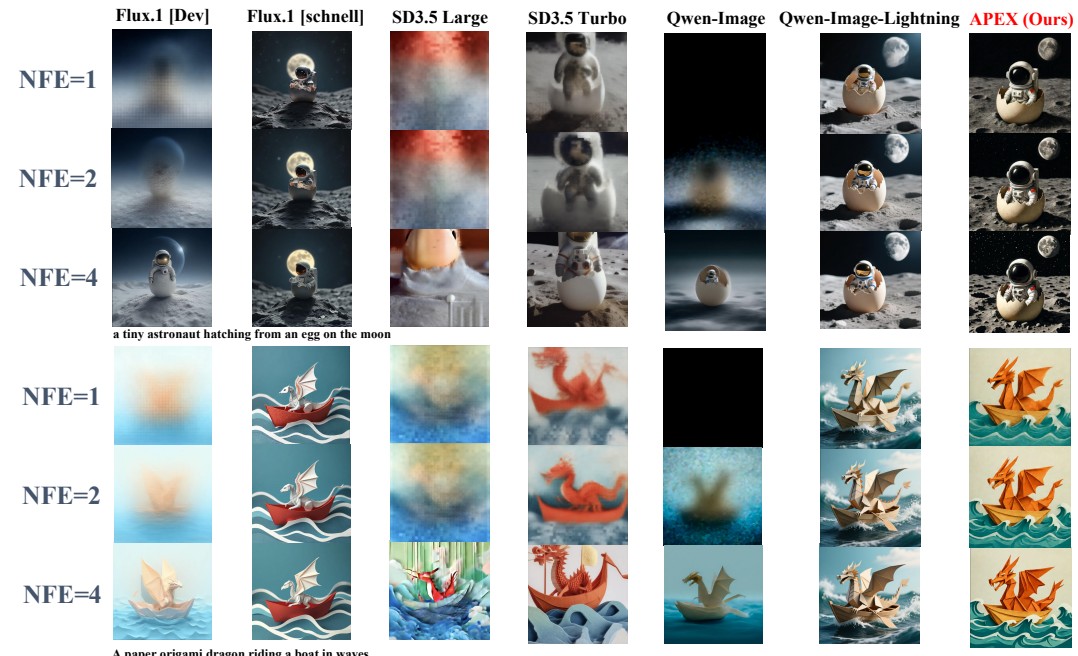

Figure 3: **Qualitative Analysis between APEX and existing methods under different NFEs.**

where $\lambda \in [0,1)$ interpolates between time steps and $\hat{\omega}(t)$ is a time-dependent weighting term. As established in prior work (Sun et al., 2025), this objective (7) is gradient-equivalent to a dual-component formulation (8) in velocity space with $\hat{\omega}(t) = t^2 \cdot \omega(t)$:

$$\mathcal{L}_{\text{path}}^{\text{dual}}(\boldsymbol{\theta}) = \mathbb{E}_{\mathbf{x},\mathbf{z},t}\left[\frac{1-\lambda}{\omega(t)}\underbrace{\|\boldsymbol{F_\theta}(\mathbf{x}_t, t) - (\mathbf{z}-\mathbf{x})\|_2^2}_{\text{Multi-step Supervision (Flow Matching)}} + \frac{\lambda}{\omega(t)}\underbrace{\|\boldsymbol{F_\theta}(\mathbf{x}_t, t) - \boldsymbol{F_{\theta^-}}(\mathbf{x}_{\lambda t}, \lambda t)\|_2^2}_{\text{Self Alignment Regularization}}\right]. \quad (8)$$

This dual-component formulation reveals the objective's hybrid nature: it simultaneously encourages the model to match the ground-truth velocity field (akin to flow matching) and to maintain consistent velocity predictions along its own generated trajectories (akin to consistency models). Under the widely used linear transport $\alpha(t) = t$, $\gamma(t) = 1 - t$, the endpoint map reduces to

$$f^{\mathbf{x}}(\boldsymbol{F_\theta}(\mathbf{x}_t, t), \mathbf{x}_t, t) = \mathbf{x}_t - t \cdot \boldsymbol{F_\theta}(\mathbf{x}_t, t), \quad (9)$$

which makes the gradient of $\mathcal{L}_{\text{path}}^{\text{primal}}$ exactly match that of $\mathcal{L}_{\text{path}}^{\text{dual}}$. This equivalence justifies optimizing either form and motivates the curvature-regularized target below.

Instead of directly combining the two terms in the dual-component loss, APEX uses a more accurate objective that implicitly regularizes path curvature via a second-order correction term. We construct a target velocity $F_{\text{target}}$ that incorporates a numerically stabilized estimate of the path's derivative. First, we define the difference in the endpoint prediction and a related coefficient over an interval $[\lambda t, t]$:

$$\Delta f^{\mathbf{x}} := f^{\mathbf{x}}(\boldsymbol{F_{\theta^-}}(\mathbf{x}_t, t), \mathbf{x}_t, t) - f^{\mathbf{x}}(\boldsymbol{F_{\theta^-}}(\mathbf{x}_{\lambda t}, \lambda t), \mathbf{x}_{\lambda t}, \lambda t), \quad (10)$$

$$\Delta a(t) := a(t) - a(\lambda t), \quad \text{where } a(t) := \alpha(t)/(\alpha(t)\hat{\gamma}(t) - \hat{\alpha}(t)\gamma(t)). \quad (11)$$

Under the common linear transport $\alpha(t)=t$, $\gamma(t)=1-t$ and flow-matching targets $\hat{\alpha}(t)=\frac{\mathrm{d}}{\mathrm{d}t}\alpha(t)=1$, $\hat{\gamma}(t)=\frac{\mathrm{d}}{\mathrm{d}t}\gamma(t)=-1$, we have $a(t)=-t$ and $a(\lambda t)=-\lambda t$. The ratio $\frac{\Delta f^{\mathbf{x}}}{\Delta a(t)}$ then serves as a secant-based approximation of the path's derivative. To ensure stability, we clip this estimate and use it to define a corrected target for the student model:

$$F_{\text{target}}(\mathbf{x}_t, t) = \boldsymbol{F_{\theta^-}}(\mathbf{x}_t, t) - 2 \cdot \text{clip}\left(\frac{\Delta f^{\mathbf{x}}}{\Delta a(t)}, -1, 1\right). \quad (12)$$

Here the clipping is applied element-wisely to the tensor ratio $\frac{\Delta f^{\mathbf{x}}}{\Delta a(t)}$. The final path objective minimizes the distance to this curvature-aware teacher signal using a robust, multi-scale $L_2$ norm,

weighted by a time-scheduling function $\beta(t)$:

$$\mathcal{L}_{\text{path}}(\boldsymbol{\theta}) = \mathbb{E}_{(\mathbf{x},\mathbf{z})\sim p(\mathbf{x},\mathbf{z}),t}\left[\beta(t)\cdot\left\|\boldsymbol{F_\theta}(\mathbf{x}_t,t,c) - \text{sg}(F_{\text{target}}(\mathbf{x}_t,t))\right\|^2_{\text{multi-}L_2}\right]. \tag{13}$$

This formulation penalizes sharp local changes in the predicted velocity field, effectively regularizing the path's local curvature and promoting smoother trajectories that are more amenable to large-step, single-call integration. Concretely, we instantiate the multi-scale norm as

$$\left\|E\right\|^2_{\text{multi-}L_2} := \tfrac{1}{3}\left(\mathcal{L}_{\text{adapt}}(E) + \mathcal{L}_{\text{adapt}}(\mathcal{D}_2(E)) + \mathcal{L}_{\text{adapt}}(\mathcal{D}_4(E))\right), \tag{14}$$

where $\mathcal{D}_s(\cdot)$ denotes spatial downsampling by factor $s$ using area interpolation, and $\mathcal{L}_{\text{adapt}}(E) = \frac{\|E\|^2_2}{(\|E\|^2_2+c)^p}$ is an adaptive loss function (Barron, 2019) that improves training stability by down-weighting large errors. We use $p = 0.5$ and $c = 10^{-3}$. We also write $\boldsymbol{F_\theta}(\mathbf{x}_t, t, c)$ when an external conditioning $c$ (e.g., text embeddings) is present. Intuitively, the secant ratio $\frac{\Delta f^x}{\Delta a(t)}$ approximates a local derivative of the endpoint map with respect to the scalar $a$, so subtracting a clipped version from the teacher velocity acts as a curvature-aware correction while preserving scale.

## 3.2 ENDPOINT FIDELITY OBJECTIVE

While $\mathcal{L}_{\text{path}}$ ensures numerical stability, it does not, by itself, guarantee that the integrated path terminates on the manifold of photorealistic images. To achieve this, APEX introduces $\mathcal{L}_{\text{endpoint}}$, a discriminator-free adversarial objective based on a novel *Self-Condition Shifting* mechanism, which is activated in the few-step generation regime where consistency is maximized (i.e., as $\lambda \to 1$).

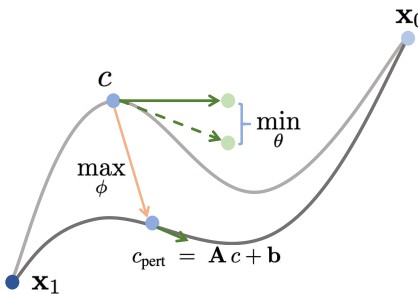

Figure 4: **Self Condition Shifting.**

**Mechanism.** The core idea is to generate an endogenous adversarial signal by contrasting the model's outputs under a strategically perturbed condition. First, we synthesize a controllable adversarial datapoint $\mathbf{x}_t^{\text{adv}}$ on the transport path by sampling a pseudo-target $\mathbf{x}_{\text{fake}}$ from a buffer of recent generations and a noise vector $\mathbf{z}_{\text{fake}} \sim \mathcal{N}(0, I)$:

$$\mathbf{x}_t^{\text{adv}} = \alpha(t)\cdot\mathbf{z}_{\text{fake}} + \gamma(t)\cdot\mathbf{x}_{\text{fake}}. \tag{15}$$

Next, we generate a single perturbed condition from the original condition $c$ via a linear transformation:

$$c_{\text{pert}} = \mathbf{A}\,c + \mathbf{b}, \tag{16}$$

where $\mathbf{A}$ is a scaling matrix and $\mathbf{b}$ is a bias vector. This allows us to define the two synergistic components of the endpoint objective.

**Objective.** The first component provides an absolute anchor to the data manifold by directly regressing the true velocity field under the perturbed condition $c_{\text{pert}}$:

$$\mathcal{L}_{\text{adv}}(\boldsymbol{\theta}) = \mathbb{E}\big[\|\boldsymbol{F_\theta}(\mathbf{x}_t^{\text{adv}}, t, c_{\text{pert}}) - (\mathbf{z}_{\text{fake}} - \mathbf{x}_{\text{fake}})\|^2_{\text{multi-}L_2}\big]. \tag{17}$$

This serves as a discriminator-free adversarial loss, directly penalizing deviations from the correct endpoint direction. The second component enforces robustness via self-consistency regularization. We first define an internal adversarial residual from the model's outputs under the perturbed and original conditions:

$$\Delta\mathbf{d} := \boldsymbol{F_\theta}(\mathbf{x}_t^{\text{adv}}, t, c_{\text{pert}}) - \text{sg}(\boldsymbol{F_{\theta^-}}(\mathbf{x}_t^{\text{adv}}, t, c)). \tag{18}$$

We then regularize a reference prediction by injecting this residual. Concretely, we define an anchor field by negating the student's prediction on a random subset of the batch, $\mathbf{F}_{\text{anchor}} = -\boldsymbol{F_\theta}(\mathbf{x}_t, t, c)$. The shifting consistency loss is then defined as:

$$\mathcal{L}_{\text{shift}}(\boldsymbol{\theta}) = \mathbb{E}\Big[\big\|\mathbf{F}_{\text{anchor}} - \text{sg}\big(\mathbf{F}_{\text{anchor}} + \Delta\mathbf{d}\big)\big\|^2_{\text{multi-}L_2}\Big]. \tag{19}$$

Together, these two components form the endpoint objective, $\mathcal{L}_{\text{endpoint}} = \mathcal{L}_{\text{adv}} + \mathcal{L}_{\text{shift}}$, which enforces both realism and robustness without requiring an external discriminator.

Table 1: **System-level comparison of efficiency and quality.** Speeds are on a single A100 (**BF16**). Throughput is samples/s (batch=10); latency is seconds (batch=1). **GenEval** is the *primary* quality metric; FID/CLIP are reported for completeness. The **best** and second-best entries are highlighted. † indicates methods requiring distinct models per NFE. **Notation:** Blue=full tuning; Red=LoRA; X.B=trainable params (B); $r$=LoRA rank.

| | Methods | NFEs | Throughput (samples/s) | Latency (s) | Params (B) | FID ↓ | CLIP ↑ | GenEval ↑ |
|---|---|---|---|---|---|---|---|---|
| *Few-Step Distillation Models* | SDXL-LCM Luo et al. (2023) | 2 | 2.89 | 0.40 | 0.9 | 18.11 | 27.51 | 0.44 |
| | PixArt-LCM Chen et al. (2024c) | 2 | 3.52 | 0.31 | 0.6 | 10.33 | 27.24 | 0.42 |
| | SD3.5-Turbo Esser et al. (2024) | 2 | 1.61 | 0.68 | 8.0 | 51.47 | 25.59 | 0.53 |
| | PCM Wang et al. (2024a)† | 2 | 2.62 | 0.56 | 0.9 | 14.70 | 27.66 | 0.55 |
| | SDXL-DMD2 Yin et al. (2024)† | 2 | 2.89 | 0.40 | 0.9 | *7.61* | 28.87 | 0.58 |
| | FLUX-schnell (Labs, 2024) | 2 | 0.92 | 1.15 | 12.0 | 7.75 | 28.25 | 0.71 |
| | Sana-Sprint (Chen et al., 2025b) | 2 | 6.46 | 0.25 | 0.6 | 6.54 | *28.40* | 0.76 |
| | Sana-Sprint (Chen et al., 2025b) | 2 | 5.68 | 0.24 | 1.6 | 6.50 | 28.45 | 0.77 |
| | Qwen-Image-Lightning (ModelTC, 2025) | 2 | 3.15 | 0.48 | 20 (r=64,0.4) | - | - | 0.85 |
| | **APEX** | 2 | 6.50 | 0.25 | 0.6 | 6.75 | *28.33* | 0.84 |
| | **APEX** | 2 | 5.72 | 0.23 | 1.6 | 6.42 | *28.24* | 0.85 |
| | **APEX** | 2 | 3.21 | 0.49 | 20 (r=32,0.2) | 6.72 | *28.71* | 0.87 |
| | **APEX** | 2 | 3.17 | 0.47 | 20 (r=64,0.4) | 6.51 | *28.42* | 0.89 |
| | **APEX** | 2 | 3.30 | 0.45 | 20 | - | - | **0.90** |
| | SDXL-LCM Luo et al. (2023) | 1 | 3.36 | 0.32 | 0.9 | 50.51 | 24.45 | 0.28 |
| | PixArt-LCM Chen et al. (2024c) | 1 | 4.26 | 0.25 | 0.6 | 73.35 | 23.99 | 0.41 |
| | PixArt-DMD Chen et al. (2024b)† | 1 | 4.26 | 0.25 | 0.6 | 9.59 | 26.98 | 0.45 |
| | SD3.5-Turbo Esser et al. (2024) | 1 | 2.48 | 0.45 | 8.0 | 52.40 | 25.40 | 0.51 |
| | PCM Wang et al. (2024a)† | 1 | 3.16 | 0.40 | 0.9 | 30.11 | 26.47 | 0.42 |
| | SDXL-DMD2 Yin et al. (2024)† | 1 | 3.36 | 0.32 | 0.9 | 7.10 | 28.93 | 0.59 |
| | FLUX-schnell (Labs, 2024) | 1 | 1.58 | 0.68 | 12.0 | 7.26 | 28.49 | 0.69 |
| | Sana-Sprint (Chen et al., 2025b) | 1 | 7.22 | 0.21 | 0.6 | 7.04 | 28.04 | 0.72 |
| | Sana-Sprint (Chen et al., 2025b) | 1 | 6.71 | 0.21 | 1.6 | 7.69 | *28.27* | 0.76 |
| | Qwen-Image-Lightning (ModelTC, 2025) | 1 | 3.29 | 0.40 | 20 (r=64,0.4) | - | - | 0.85 |
| | MeanFlow (Geng et al., 2025) | 1 | 3.29 | 0.40 | 20 | - | - | 0.05 |
| | CM (Song et al., 2023) | 1 | 3.29 | 0.40 | 20 | - | - | 0.01 |
| | CTM (Lu & et al., 2024) | 1 | 3.29 | 0.40 | 20 | - | - | 0.10 |
| | UCGM (Sun et al., 2025) | 1 | 3.29 | 0.40 | 20 | - | - | 0.43 |
| | **APEX** | 1 | 7.30 | 0.20 | 0.6 | 6.99 | *28.36* | **0.84** |
| | **APEX** | 1 | 6.84 | 0.20 | 1.6 | 6.78 | *28.12* | **0.84** |
| | **APEX** | 1 | 3.29 | 0.39 | 20 (r=32,0.2) | 7.22 | *28.62* | 0.88 |
| | **APEX** | 1 | 3.27 | 0.39 | 20 (r=64,0.4) | 7.14 | *28.45* | 0.89 |
| | **APEX** | 1 | 3.50 | 0.34 | 20 | - | - | **0.89** |

## 3.3 ANY-STEP SAMPLING

The APEX framework naturally supports flexible, any-step sampling. Because the model is trained with objectives that ensure both path integrability and endpoint fidelity, it is robust to a wide range of NFEs, from one-step synthesis to many-step iterative refinement. The core sampling loop reconstructs the next state $\mathbf{x}_{t_{i+1}}$ from the current state's predicted clean data $\hat{\mathbf{x}}_i$ and noise $\hat{\mathbf{z}}_i$:

$$\mathbf{x}_{t_{i+1}} = \gamma(t_{i+1}) \cdot \hat{\mathbf{x}}_i + \alpha(t_{i+1}) \cdot \hat{\mathbf{z}}_i \tag{20}$$

To further accelerate convergence and improve quality, we employ an estimation extrapolation technique. This uses a momentum-like term to refine the current prediction based on the previous one, effectively canceling out systematic error terms in the estimation:

$$\hat{\mathbf{x}}_i \leftarrow \hat{\mathbf{x}}_i + \kappa \cdot \left( \hat{\mathbf{x}}_i - \hat{\mathbf{x}}_{i-1} \right) \tag{21}$$

where $\kappa$ is the extrapolation ratio. This allows for higher-quality results with fewer sampling steps, solidifying APEX's efficiency.

## 4 EXPERIMENTS

### 4.1 EXPERIMENTAL SETUP

• ***Backbones and tuning.*** We consider three capacities: APEX 0.6B and APEX 1.6B (full-parameter tuning), and APEX 20B using LoRA on Qwen-Image (Wu et al., 2025a).

• ***Datasets.*** Our training data comprises both open-source and newly synthesized datasets. We utilize ShareGPT-4o(Chen et al., 2025c) and BLIP-3o(Chen et al., 2025a) as our primary open-source resources. Additionally, we construct two synthetic datasets using the Qwen-Image-20B model. These include 50K samples generated from prompts in the Flux-Reasoning-6M dataset (Fang et al., 2025), and another 50K samples synthesized from BLIP-3o prompts.

Table 2: Quantitative Evaluation results on GenEval.

| Model | Single Object | Two Object | Counting | Colors | Position | Attribute Binding | Overall↑ |
|---|---|---|---|---|---|---|---|
| Show-o (Xie et al., 2024b) | 0.95 | 0.52 | 0.49 | 0.82 | 0.11 | 0.28 | 0.53 |
| Emu3-Gen (Wang et al., 2024b) | 0.98 | 0.71 | 0.34 | 0.81 | 0.17 | 0.21 | 0.54 |
| PixArt-$\alpha$ (Chen et al., 2024d) | 0.98 | 0.50 | 0.44 | 0.80 | 0.08 | 0.07 | 0.48 |
| SD3 Medium (Esser et al., 2024) | 0.98 | 0.74 | 0.63 | 0.67 | 0.34 | 0.36 | 0.62 |
| FLUX.1 [Dev] (BlackForest, 2024) | 0.98 | 0.81 | 0.74 | 0.79 | 0.22 | 0.45 | 0.66 |
| SD3.5 Large (Esser et al., 2024) | 0.98 | 0.89 | 0.73 | 0.83 | 0.34 | 0.47 | 0.71 |
| JanusFlow (Ma et al., 2025) | 0.97 | 0.59 | 0.45 | 0.83 | 0.53 | 0.42 | 0.63 |
| Lumina-Image 2.0 (Qin et al., 2025) | - | 0.87 | 0.67 | - | - | 0.62 | 0.73 |
| Janus-Pro-7B (Chen et al., 2025d) | 0.99 | 0.89 | 0.59 | 0.90 | 0.79 | 0.66 | 0.80 |
| HiDream-I1-Full (Cai et al., 2025) | **1.00** | 0.98 | 0.79 | 0.91 | 0.60 | 0.72 | 0.83 |
| GPT Image 1 [High] (OpenAI, 2025) | 0.99 | 0.92 | 0.85 | 0.92 | 0.75 | 0.61 | 0.84 |
| Seedream 3.0 (Gao et al., 2025) | 0.99 | **0.96** | 0.91 | **0.93** | 0.47 | 0.80 | 0.84 |
| BAGEL (Deng et al., 2025) | 0.98 | 0.95 | 0.84 | 0.95 | 0.78 | 0.77 | 0.88 |
| Qwen-Image (Wu et al., 2025a) | 0.99 | 0.92 | 0.89 | 0.88 | 0.76 | 0.77 | 0.87 |
| Hyper-BAGEL (Lu et al., 2025) | 0.97 | 0.86 | 0.75 | 0.90 | 0.67 | 0.62 | 0.80 |
| Qwen-Image-RL (Wu et al., 2025a) | **1.00** | 0.95 | **0.93** | 0.92 | **0.87** | **0.83** | **0.91** |
| Qwen-Image-Lightning (ModelTC, 2025) | - | - | - | - | - | - | 0.85 |
| **APEX 0.6B (1-NFE)** | 0.99 | 0.91 | 0.75 | **0.93** | 0.76 | 0.69 | 0.84 |
| **APEX 1.6B (1-NFE)** | 0.99 | 0.91 | 0.75 | **0.93** | 0.76 | 0.68 | 0.84 |
| **APEX 20B (LoRA&r=32) (1-NFE)** | 0.99 | 0.95 | 0.85 | 0.90 | 0.79 | 0.78 | 0.88 |
| **APEX 20B (LoRA&r=64) (1-NFE)** | 0.99 | 0.94 | 0.88 | 0.90 | **0.85** | 0.78 | **0.89** |
| **APEX 20B (SFT) (1-NFE)** | 0.99 | 0.92 | 0.83 | 0.91 | 0.86 | 0.81 | **0.89** |

Table 3: Quantitative evaluation results on DPGBench.

| Model | Global | Entity | Attribute | Relation | Other | Overall↑ |
|---|---|---|---|---|---|---|
| SD v1.5 (Rombach et al., 2022) | 74.63 | 74.23 | 75.39 | 73.49 | 67.81 | 63.18 |
| PixArt-$\alpha$ (Chen et al., 2024d) | 74.97 | 79.32 | 78.60 | 82.57 | 76.96 | 71.11 |
| Lumina-Next (Zhuo et al., 2024) | 82.82 | 88.65 | 86.44 | 80.53 | 81.82 | 74.63 |
| SDXL (Podell et al., 2023) | 83.27 | 82.43 | 80.91 | 86.76 | 80.41 | 74.65 |
| Hunyuan-DiT (Li et al., 2024b) | 84.59 | 80.59 | 88.01 | 74.36 | 86.41 | 78.87 |
| Janus (Wu et al., 2025b) | 82.33 | 87.38 | 87.70 | 85.46 | 86.41 | 79.68 |
| PixArt-$\Sigma$ (Chen et al., 2024a) | 86.89 | 82.89 | 88.94 | 86.59 | 87.68 | 80.54 |
| Emu3-Gen (Wang et al., 2024b) | 85.21 | 86.68 | 86.84 | 90.22 | 83.15 | 80.60 |
| Janus-Pro-1B (Chen et al., 2025d) | 87.58 | 88.63 | 88.17 | 88.98 | 88.30 | 82.63 |
| DALL-E 3 (OpenAI, 2023) | 90.97 | 89.61 | 88.39 | 90.58 | 89.83 | 83.50 |
| FLUX.1 [Dev] (BlackForest, 2024) | 74.35 | 90.00 | 88.96 | 90.87 | 88.33 | 83.84 |
| SD3.5-Medium Esser et al. (2024) | 84.08 | 87.90 | 91.01 | 88.83 | 80.70 | 88.68 |
| SD3.5-Turbo Sauer et al. (2024b) | 79.03 | 80.12 | 86.13 | 84.73 | 91.86 | 78.29 |
| SD3.5-Large Esser et al. (2024) | 83.21 | 84.27 | 88.99 | 87.35 | 93.28 | 80.35 |
| FLUX.1-schnell Labs (2024) | 84.94 | 86.62 | 90.82 | 88.35 | 93.45 | 82.00 |
| Janus-Pro-7B (Chen et al., 2025d) | 86.90 | 88.90 | 89.40 | 89.32 | 89.48 | 84.19 |
| HiDream-I1-Full (Cai et al., 2025) | 76.44 | 90.22 | 89.48 | 93.74 | 91.83 | 85.89 |
| Lumina-Image 2.0 (Qin et al., 2025) | - | 91.97 | 90.20 | **94.85** | - | 87.20 |
| Seedream 3.0 (Gao et al., 2025) | **94.31** | **92.65** | 91.36 | 92.78 | 88.24 | 88.27 |
| GPT Image 1 [High] (OpenAI, 2025) | 88.89 | 88.94 | 89.84 | 92.63 | 90.96 | 85.15 |
| Qwen-Image (Wu et al., 2025a) | 91.32 | 91.56 | **92.02** | 94.31 | **92.73** | 88.32 |
| Playground v3 (Liu et al., 2024) | 87.04 | 91.94 | 85.71 | 90.90 | 90.00 | **92.72** |
| Qwen-Image-Lightning (ModelTC, 2025) | - | - | - | - | - | 87.79 |
| **APEX 0.6B (1-NFE)** | 90.58 | 90.36 | 90.44 | 90.77 | 90.73 | 82.66 |
| **APEX 1.6B (1-NFE)** | 90.77 | 90.56 | 90.63 | 90.98 | 90.94 | 83.22 |
| **APEX 20B (LoRA&r=32) (1-NFE)** | 93.12 | 90.95 | 91.38 | 90.65 | 91.73 | 86.17 |
| **APEX 20B (LoRA&r=64) (1-NFE)** | 92.46 | 91.14 | 90.71 | 91.30 | 91.98 | 85.77 |
| **APEX 20B (SFT) (1-NFE)** | 93.25 | 89.76 | 90.65 | 91.17 | 90.75 | 84.59 |

• *Training and hardware.* Training uses BF16 precision. For LoRA, we vary the rank $r \in \{32, 64\}$ and keep all other settings identical across ranks. We use 16×NVIDIA H100 80GB, 8×NVIDIA H800 80GB, 8×A100 80GB GPUs for training and evaluation.

• *Evaluation metrics.* Our *primary* metric is GenEval Overall (Ghosh et al., 2023). We also report FID and CLIP on MJHQ-30K (Li et al., 2024a), DPGBench (Hu et al., 2024) and WISE (Niu et al., 2025) for completeness. Unless noted, results are with NFE=1.

## 4.2 Efficiency and Performance Comparison

We profile APEX under **NFE=1/2** and contrast it with the strongest prior distilled model at each setting, as summarized in Table 1. **GenEval Overall** is our headline metric, with throughput and

Table 4: **Quantitative evaluation results on WISE.**

| Model | Cultural | Time | Space | Biology | Physics | Chemistry | Overall↑ |
|---|---|---|---|---|---|---|---|
| SD v1.5 (Rombach et al., 2022) | 0.34 | 0.35 | 0.32 | 0.28 | 0.29 | 0.21 | 0.32 |
| SDXL (Podell et al., 2023) | 0.43 | 0.48 | 0.47 | 0.44 | 0.45 | 0.27 | 0.43 |
| SD3.5-Large Esser et al. (2024) | 0.44 | 0.50 | 0.58 | 0.44 | 0.52 | 0.31 | 0.46 |
| PixArt-$\alpha$ (Chen et al., 2024d) | 0.45 | 0.50 | 0.48 | 0.49 | 0.56 | 0.34 | 0.47 |
| Playground-v2.5 (Li et al., 2024a) | 0.49 | 0.58 | 0.55 | 0.43 | 0.48 | 0.33 | 0.49 |
| FLUX.1 [Dev] (BlackForest, 2024) | 0.48 | 0.58 | 0.62 | 0.42 | 0.51 | 0.35 | 0.50 |
| Janus (Wu et al., 2025b) | 0.16 | 0.26 | 0.35 | 0.28 | 0.30 | 0.14 | 0.23 |
| VILA-U (Wu et al., 2024) | 0.51 | 0.51 | 0.51 | 0.49 | 0.51 | 0.49 | 0.50 |
| Show-o (Xie et al., 2024b) | 0.95 | 0.52 | 0.49 | 0.82 | 0.11 | 0.28 | 0.53 |
| Janus-Pro-7B (Chen et al., 2025d) | 0.30 | 0.37 | 0.49 | 0.36 | 0.42 | 0.26 | 0.35 |
| Emu3-Gen (Wang et al., 2024b) | 0.34 | 0.45 | 0.48 | 0.41 | 0.45 | 0.47 | 0.39 |
| MetaQuery-XL (Pan et al., 2025) | 0.56 | 0.55 | 0.62 | 0.49 | 0.63 | 0.41 | 0.55 |
| BAGEL (Deng et al., 2025) | 0.44 | 0.55 | 0.68 | 0.44 | 0.60 | 0.39 | 0.52 |
| GPT-4o | 0.81 | 0.71 | 0.89 | 0.83 | 0.79 | 0.74 | **0.80** |
| Qwen-Image (Wu et al., 2025a) | - | - | - | - | - | - | 0.62 |
| Qwen-Image-Lightning (ModelTC, 2025) | - | - | - | - | - | - | 0.51 |
| **APEX 20B (SFT) (1-NFE)** | 0.53 | 0.54 | 0.66 | 0.48 | 0.61 | 0.41 | 0.54 |

Table 5: **Effect of training data and steps on GenEval Overall (NFE=1).** We compare **ShareGPT-4o** and **BLIP-3o** across training steps for APEX 0.6B/1.6B, and LoRA-tuned Qwen-Image 20B with ranks r=32/r=64. All runs use global batch size **64** and **BF16**.

| Model | ShareGPT-4o | | | Blip-3o | | |
|---|---|---|---|---|---|---|
| | 2Ksteps | 8Ksteps | 10Ksteps | 2Ksteps | 8Ksteps | 10Ksteps |
| APEX 0.6B | 0.37 | 0.67 | 0.73 | 0.71 | 0.77 | 0.81 |
| APEX 1.6B | 0.36 | 0.70 | 0.73 | 0.27 | 0.78 | 0.83 |
| | 0.4Ksteps | 1Ksteps | 2Ksteps | 0.4Ksteps | 1Ksteps | 2Ksteps |
| APEX 20B (r=32) | 0.19 | 0.33 | 0.62 | 0.74 | 0.84 | 0.83 |
| APEX 20B (r=64) | 0.21 | 0.35 | 0.61 | 0.73 | 0.85 | 0.84 |

latency reported to highlight practical applicability. At **NFE=1**, APEX 0.6B sustains **7.3** samples/s at **0.20**s latency while achieving **0.84** GenEval—a $\approx 0.15$ absolute gain over the best previously published one-step baseline (FLUX-schnell at 0.69) despite using roughly twenty times fewer parameters. Scaling to APEX 1.6B keeps latency flat with similar throughput. Our LoRA-tuned APEX 20B further lifts GenEval to **0.89** (r=64) at only **0.39**s latency, which is state of art. Moving to **NFE=2**, APEX 1.6B rises to **0.85** GenEval, a $\sim 8$ point margin over the strongest two-step baseline (Sana-Sprint 1.6B at 0.77) while running more than twice as fast. The 20B LoRA variant sustains **0.89** GenEval with a modest latency bump to **0.47**s, suggesting that the higher-order self-consistency keeps the transport stable even as we introduce one extra refinement step. Taken together, these results show that APEX closes the quality gap to multi-step generators without giving up the latency headroom that makes distilled models attractive in production pipelines.

### 4.3 ABLATIONS

We present controlled ablations to isolate the effects of key design choices in APEX. Unless otherwise stated, all results are reported with **NFE=1** and the **GenEval Overall** metric, using identical prompts, seeds abd resolution.

**Balancing $\mathcal{L}_{\text{path}}$ and $\mathcal{L}_{\text{endpoint}}$.** We dissect the contribution of the path-integrability objective and the endpoint-fidelity objective by ablating weights $\lambda_p : \lambda_e$ in $\mathcal{L} = \lambda_p \mathcal{L}_{\text{path}} + \lambda_e \mathcal{L}_{\text{endpoint}}$ on three models: APEX 0.6B, 1.6B, and 20B (LoRA). As shown in Table 7, either component alone underperforms the balanced settings. A mild endpoint emphasis (e.g., 1.0:0.5) or equal weighting (1.0:1.0) yields the highest GenEval, whereas excessive endpoint emphasis (1.0:2.0) slightly harms integrability and overall score. This validates our premise: path regularization is necessary to retain one-step stability, whereas the endpoint term is critical to reach high-fidelity outputs.
• *Condition shifting hyperparameters $a$ and $b$.* To probe the self-conditioned contrast, we vary the scale $a$ and bias $b$ in $c_{\text{pert}} = \mathbf{A}\,c + \mathbf{b}$ and report GenEval on a $(a, b)$ grid

Table 6: **Effect of condition-shifting hyperparameters on GenEval Overall (NFE=1).** We vary the linear transform $c_{\text{pert}} = \mathbf{A}\,c + \mathbf{b}$ with $\mathbf{A} = a\,\mathbf{I}$ (scalar scale) and a uniform bias $b$, and report **Overall** on a grid over $(a, b)$. Results use **APEX 20B (LoRA, r=32)**, **BLIP-3o** data, and **0.4K** training steps. Moderate negative scaling ($a \in \{-1.0, -0.5\}$) with small positive bias ($b \in \{0.1, 1.0\}$) yields the most robust gains.

| $a \setminus b$ | 0.0 | 0.1 | 1.0 | 10.0 |
|---|---|---|---|---|
| $-1.0$ | 0.76 | 0.73 | 0.74 | 0.74 |
| $-0.5$ | 0.75 | 0.79 | 0.81 | 0.70 |
| $0.5$ | 0.29 | 0.37 | 0.30 | 0.73 |

Table 7: **Ablation on the weights of $\mathcal{L}_{\text{path}}$ vs $\mathcal{L}_{\text{endpoint}}$.** We report GenEval Overall (NFE=1) for different weighting ratios ($\lambda_p : \lambda_e$). The dataset is BLIP-3o. Training steps are 8K for 0.6B/1.6B models and 0.4K for the 20B (LoRA) model. Best per model in **bold**.

| Weighting Ratio ($\lambda_p : \lambda_e$) | APEX 0.6B | APEX 1.6B | APEX 20B ($r=32$) |
|---|---|---|---|
| 1.0 : 0.0 (Path Only) | 0.32 | 0.35 | 0.42 |
| 0.0 : 1.0 (Endpoint Only) | 0.63 | 0.66 | 0.69 |
| 1.0 : 0.5 | 0.72 | 0.71 | 0.81 |
| **1.0 : 1.0 (Ours)** | **0.77** | **0.76** | **0.83** |
| 1.0 : 2.0 | 0.74 | 0.75 | 0.82 |

Table 6 . Results indicate a broad optimum around $a \in \{-1.0, -0.5\}$ with small positive biases ($b \in [0.1, 1.0]$), aligning with our design choice of moderate scaling and light shifts. Positive scaling ($a=0.5$) is generally suboptimal unless paired with a larger bias ($b=10.0$).

• *Datasets vs. training steps.* We first study data and compute scaling by varying one factor at a time. The dataset ablation Table 5 compares ShareGPT-4o and BLIP-3o across fixed steps, evaluated on APEX 0.6B and 1.6B, and extends to Qwen-Image 20B (LoRA) at shorter step budgets. BLIP-3o consistently yields higher GenEval at larger step counts for both 0.6B and 1.6B (e.g., **0.81/0.83** vs **0.73** at 10K). For the 20B LoRA model, BLIP-3o reaches **0.84–0.85** by 1–2K steps, whereas ShareGPT-4o improves steadily with more steps (**0.19 → 0.62**).

## 5 CONCLUSION AND FUTURE WORK

In this work, we studied the core tension between *path integrability* and *endpoint fidelity* that limits few-step generative modeling. We presented **APEX**, which resolves this tension via two synergistic components: a higher-order path self-consistency objective that directly regularizes local curvature of the transport, and a discriminator-free *self-condition shifting* mechanism that anchors the endpoint to the data manifold while improving perceptual realism. APEX attains state-of-the-art few-step quality without sacrificing speed. At **NFE=1**, the 0.6B/1.6B models reach **0.84** GenEval with **0.20**s latency (**7.3/6.84** samples/s), and the 20B LoRA variant achieves **0.89** GenEval at **0.39**s latency. The 20B LoRA model sustains **0.89** GenEval at **0.47**s latency. These results indicate that curvature-aware path training, combined with self-conditioned endpoint alignment, closes the gap to multi-step generators in a single or very few steps.

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

CONTENTS

## A    LLM USAGE STATEMENT

LLMs were used solely as auxiliary tools for grammar checking and language polishing. They did not contribute to the generation of research ideas, the design of experiments, the development of methodologies, data analysis, or any substantive aspects of the research. All scientific content, conceptual contributions, and experimental results are entirely the work of the authors. The authors take full responsibility for the contents of this paper.

## B    LIMITATIONS

**Fairness of Comparisons in control setting.**    While we have made every effort to benchmark APEX against state-of-the-art methods under comparable conditions, achieving perfectly controlled comparisons remains a significant challenge in the rapidly evolving field of generative models. Minor discrepancies in training datasets, subtle architectural variations in backbones, and undisclosed implementation details across different studies can influence final performance metrics.

**Scope of Application.**    The experiments in this work are primarily focused on demonstrating the effectiveness of APEX for high-fidelity text-to-image synthesis. We have not extended our evaluation to other important generative tasks such as super-resolution, image in-painting, or video generation. While we believe the core principles of ensuring path integrability and endpoint fidelity are generalizable, adapting and validating APEX for these domains requires further research and engineering effort, which we leave as a promising direction for future work.

**Engineering Optimizations.**    The efficiency and speed reported in our experiments are direct results of the algorithmic design of APEX, which enables high-quality synthesis in a single inference step. Our implementation does not yet incorporate advanced engineering optimizations that could further accelerate inference. Techniques such as model quantization, operator fusion, and deployment with specialized inference engines are orthogonal to our core contribution but offer a practical path to achieving even lower latency in production environments.

## C    RELATED WORK

### C.1    FROM MACRO-LEVEL TO LOCAL CONTROL

The foundational paradigm in continuous generative modeling, including diffusion (Ho et al., 2020; Song et al., 2020; Karras et al., 2022a) and flow-matching (Lipman et al., 2022), involves learning an *instantaneous* velocity field. While effective for multi-step integration, this first-order approach is brittle under coarse discretization, as high path curvature causes truncation errors that degrade few-step generation quality (Karras et al., 2022a).

To address this, a significant body of work has shifted focus from instantaneous dynamics to supervising the model's behavior over a *time interval*. These methods attempt to ensure path integrability at a macro level. For instance, Consistency Models (CMs) (Song et al., 2023; Lu & et al., 2024) enforce a *relative* constraint, requiring that endpoint predictions remain consistent across different points on the same trajectory. While effective, this does not directly address the geometric properties of the path that cause discretization errors. More recent approaches such as MeanFlow (Geng et al., 2025) and Transition Models (TiM) (Wang et al., 2025) go a step further by directly modeling the average velocity or state transition over an interval. They learn the *result* of a large step, but the constraint remains on the interval's endpoints rather than its internal geometry.

UCGM (Sun et al., 2025) unifies different paradigms by interpolating between their respective training objectives with a hyperparameter. APEX adopts a more fundamental approach. Instead of correcting the symptoms of poor integrability (i.e., inconsistent endpoints), our path loss, $\mathcal{L}_{\text{path}}$, addresses the root cause: *local path curvature*. By directly minimizing the leading-order source of discretization error via a local secant–tangent consistency objective, APEX provides a more geometrically grounded mechanism to ensure the learned path is smoothly integrable.

### C.2    FROM EXTERNAL DISCRIMINATORS TO ENDOGENOUS ADVERSARIAL

Achieving high one-step fidelity requires strong, absolute anchoring of the endpoint prediction to the data manifold, a property that relative consistency constraints alone do not guarantee. A primary approach involves incorporating external adversarial signals. Distillation methods like DMD/DMD2 (Yin et al., 2024) and other GAN-based refiners (Kim et al., 2023a; Sauer et al., 2024a;

Zheng et al., 2025) use an auxiliary discriminator to sharpen outputs, even allowing a student to surpass its teacher. However, this reliance is a double-edged sword: it introduces training instability, computational overhead, and, critically, often depends on a costly pre-computed dataset for regularization. For large-scale models, generating this dataset of teacher-student pairs can be prohibitively expensive, exceeding the cost of training itself (Yin et al., 2024). A distinct line of work generates adversarial signals internally. Direct Discriminative Optimization (DDO) (Zheng et al., 2025) re-parameterizes the GAN discriminator using the likelihood ratio between a target model and a fixed reference, operating in *probability space*.

APEX advances this line by replacing external discriminators with an *endogenous adversarial signal* derived from condition shifting, while simultaneously controlling local path curvature via second-order self-consistency. This combination yields one-step fidelity without discriminator overhead or precomputed teacher datasets.

### C.3 SCALABLE TRAINING

The practical implementation of generative models, including APEX, hinges on scalable system design. A key challenge is the need to compute time derivatives to enforce interval consistency. Methods like MeanFlow (Geng et al., 2025) relied on Jacobian-Vector Products (JVP), creating a significant scalability bottleneck. JVP is computationally intensive and, more importantly, incompatible with critical training optimizations like FlashAttention (Dao, 2023) and FSDP-based distributed training (Zhao et al., 2023), limiting its use in billion-parameter models.

To overcome this, the field has converged on finite-difference estimators, often termed Differential Derivation Equations (DDE), as a scalable alternative (Lu & et al., 2024; Wang et al., 2025). These estimators rely only on forward passes and are natively compatible with modern training infrastructure. APEX's path integrability objective fully embraces this scalable approach. This design choice, combined with our efficient endogenous adversarial mechanism and established best practices for large-scale training——ensures that APEX maintains 1-NFE fidelity and any-step scaling on large backbones like SDXL, SANA, and Qwen-Image (Podell et al., 2023; Xie et al., 2024a; Wu et al., 2025a), while remaining fully compatible with parameter-efficient tuning.

## D PSEUDOCODE

### D.1 TRAINING ALGORITHM FOR APEX

---

**Algorithm 1** The APEX Training Algorithm

---

**Require:** Model $\boldsymbol{F_\theta}$, EMA model $\boldsymbol{F_{\theta^-}}$, dataset $D$, loss weights $\lambda_p, \lambda_e$.
**Ensure:** Trained model $\boldsymbol{F_\theta}$.
 1: **repeat**
 2:     Sample $(\mathbf{x}, c) \sim D$, $\mathbf{z} \sim \mathcal{N}(0, I)$, and time steps $t, \lambda t \in [0, 1]$.
 3:     Construct interpolated data $\mathbf{x}_t = \alpha(t)\mathbf{z} + \gamma(t)\mathbf{x}$ and $\mathbf{x}_{\lambda t} = \alpha(\lambda t)\mathbf{z} + \gamma(\lambda t)\mathbf{x}$.
 4:
 5:     # – Path Integrability Objective –
 6:     With the EMA model $\boldsymbol{F_{\theta^-}}$, compute endpoint predictions $\hat{\mathbf{x}}_t = f^{\mathbf{x}}(\boldsymbol{F_{\theta^-}}(\mathbf{x}_t, t), \mathbf{x}_t, t)$ and
        $\hat{\mathbf{x}}_{\lambda t} = f^{\mathbf{x}}(\boldsymbol{F_{\theta^-}}(\mathbf{x}_{\lambda t}, \lambda t), \mathbf{x}_{\lambda t}, \lambda t)$.
 7:     Approximate path derivative via secant: $\Delta f^{\mathbf{x}} = \hat{\mathbf{x}}_t - \hat{\mathbf{x}}_{\lambda t}$ and $\Delta a(t) = a(t) - a(\lambda t)$.
 8:     Construct curvature-aware target: $\boldsymbol{F}_{\text{target}} = \boldsymbol{F_{\theta^-}}(\mathbf{x}_t, t) - 2 \cdot \text{clip}\left(\frac{\Delta f^{\mathbf{x}}}{\Delta a(t)}, -1, 1\right)$.
 9:     Compute path loss: $\mathcal{L}_{\text{path}} = \mathbb{E}\left[\beta(t) \cdot \|\boldsymbol{F_\theta}(\mathbf{x}_t, t, c) - \text{sg}(\boldsymbol{F}_{\text{target}})\|^2_{\text{multi-}L_2}\right]$.
10:
11:     # – Endpoint Fidelity Objective (Self-Condition Shifting) –
12:     Sample a fake data point $(\mathbf{x}_{\text{fake}}, \mathbf{z}_{\text{fake}})$ from a buffer and a random time $t'$.
13:     Construct adversarial input: $\mathbf{x}_{t'}^{\text{adv}} = \alpha(t')\mathbf{z}_{\text{fake}} + \gamma(t')\mathbf{x}_{\text{fake}}$.
14:     Perturb condition: $c_{\text{pert}} = \mathbf{A}c + \mathbf{b}$.
15:     Compute adversarial loss (absolute anchor): $\mathcal{L}_{\text{adv}} = \mathbb{E}\left[\|\boldsymbol{F_\theta}(\mathbf{x}_{t'}^{\text{adv}}, t', c_{\text{pert}}) - (\mathbf{z}_{\text{fake}} - \mathbf{x}_{\text{fake}})\|^2_{\text{multi-}L_2}\right]$.
16:     Compute internal adversarial residual: $\Delta\mathbf{d} = \boldsymbol{F_\theta}(\mathbf{x}_{t'}^{\text{adv}}, t', c_{\text{pert}}) - \text{sg}(\boldsymbol{F_{\theta^-}}(\mathbf{x}_{t'}^{\text{adv}}, t', c))$.
17:     Define anchor field on a random batch subset: $\mathbf{F}_{\text{anchor}} = -\boldsymbol{F_\theta}(\mathbf{x}_t, t, c)$.
18:     Compute shifting consistency loss: $\mathcal{L}_{\text{shift}} = \mathbb{E}\left[\|\mathbf{F}_{\text{anchor}} - \text{sg}(\mathbf{F}_{\text{anchor}} + \Delta\mathbf{d})\|^2_{\text{multi-}L_2}\right]$.
19:     $\mathcal{L}_{\text{endpoint}} = \mathcal{L}_{\text{adv}} + \mathcal{L}_{\text{shift}}$.
20:
21:     # – Update –
22:     Total Loss: $\mathcal{L}_{\text{APEX}} = \lambda_p \mathcal{L}_{\text{path}} + \lambda_e \mathcal{L}_{\text{endpoint}}$.
23:     Update model parameters: $\boldsymbol{\theta} \leftarrow \text{optimizer\_step}(\boldsymbol{\theta}, \nabla_{\boldsymbol{\theta}} \mathcal{L}_{\text{APEX}})$.
24:     Update EMA parameters: $\boldsymbol{\theta}^- \leftarrow \text{update\_ema}(\boldsymbol{\theta}^-, \boldsymbol{\theta})$.
25: **until** Convergence

---

## D.2 SAMPLING ALGORITHM FOR APEX

---

**Algorithm 2** The APEX Any-Step Sampling Algorithm

---

**Require:** Initial noise $\mathbf{x}_N \sim \mathcal{N}(0, I)$, trained model $\boldsymbol{F_\theta}$, number of steps $N$, time schedule
   $\{t_i\}_{i=N}^0$, extrapolation ratio $\kappa$.
**Ensure:** Generated sample $\hat{\mathbf{x}}_0$.
 1: Let $\mathbf{x}_{t_N} = \mathbf{x}_N$. Initialize prediction history for extrapolation.
 2: **for** $i = N$ down to 1 **do**
 3:     Predict endpoint and noise from current state using Eqs. (6): $\hat{\mathbf{x}}_{i-1}, \hat{\mathbf{z}}_{i-1} = \text{predict}(\boldsymbol{F_\theta}, \mathbf{x}_{t_i}, t_i)$.
 4:     **if** $i < N$ and $\kappa > 0$ **then**
 5:         Apply estimation extrapolation: $\hat{\mathbf{x}}_{i-1} \leftarrow \hat{\mathbf{x}}_{i-1} + \kappa \cdot (\hat{\mathbf{x}}_{i-1} - \hat{\mathbf{x}}_i)$.
 6:     **end if**
 7:     **if** $i > 1$ **then**
 8:         Reconstruct next state on the path: $\mathbf{x}_{t_{i-1}} = \gamma(t_{i-1})\hat{\mathbf{x}}_{i-1} + \alpha(t_{i-1})\hat{\mathbf{z}}_{i-1}$.
 9:     **end if**
10: **end for**
11: **return** $\hat{\mathbf{x}}_0$.

---

# E   ANY-STEP SAMPLING AND TIME DISCRETIZATION

APEX supports robust sampling under an arbitrary number of function evaluations (NFEs). The core sampling map reconstructs the next state from the estimated clean data and noise components using the transport

$$\mathbf{x}_{t_{i+1}} \;=\; \gamma(t_{i+1})\,\hat{\mathbf{x}}_i \;+\; \alpha(t_{i+1})\,\hat{\mathbf{z}}_i.$$

We now describe the design choices that make this procedure stable from NFE=1 to multi-step regimes.

**Time Discretization.**   We draw a monotone grid $\{t_i\}_{i=0}^{N}$ over $[0,1]$ with optional boundary gaps and a smooth non-linear warping to emphasize informative regions. Concretely, we first form a uniform grid in $[\varepsilon_0,\, 1-\varepsilon_1]$, then apply the generalized Kumaraswamy transform

$$f_{\text{Kuma}}(t; a, b, c) \;=\; \big(1 - (1 - t^a)^b\big)^c, \quad a, b, c > 0,$$

and finally reverse time to follow the probability flow from prior to data. This warping yields a dense allocation of steps in the mid-range of the trajectory while avoiding potentially unstable endpoints.

**Second-Order Correction (Heun's Method).**   For NFE$\geq$2 sampling, we employ a predictor-corrector step to improve accuracy. Given a first-order prediction $(\hat{\mathbf{x}}_i, \hat{\mathbf{z}}_i)$ at time $t_i$, we first predict a candidate next state $\mathbf{x}_{t_{i+1}}^{\text{pred}}$. We then re-evaluate the model at this candidate state to get a corrected estimate $(\hat{\mathbf{x}}_{i+1}^{\text{corr}}, \hat{\mathbf{z}}_{i+1}^{\text{corr}})$. The final next state is formed by averaging the predictions, effectively canceling leading-order truncation errors.

**Controlled Stochasticity and Extrapolation.**   To balance diversity and fidelity, we introduce a bounded noise ratio $\rho \in [0, 1]$ to inject controlled stochasticity into the noise estimate at each step. Additionally, to mitigate systematic bias, we use an estimation extrapolation technique, which refines the current prediction based on a momentum-like term from the previous step:

$$\hat{\mathbf{x}}_i \leftarrow \hat{\mathbf{x}}_i + \kappa\,\big(\hat{\mathbf{x}}_i - \hat{\mathbf{x}}_{i-1}\big), \quad \kappa \in [0, 1].$$

This lightweight trick consistently improves quality for NFE$\leq$4 at no extra computational cost.

# F   TRAINING STABILITY

This section provides additional training stability results to complement the quantitative analysis in the main paper.

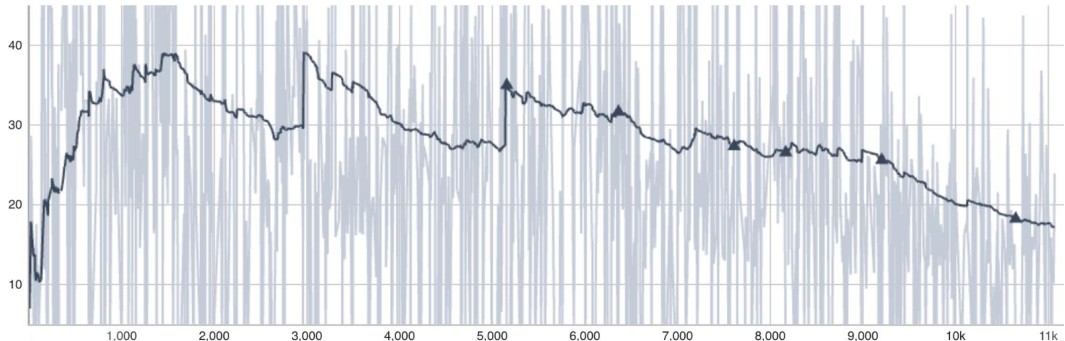

Figure 5: **Gradient Norm of APEX SFT Training.**

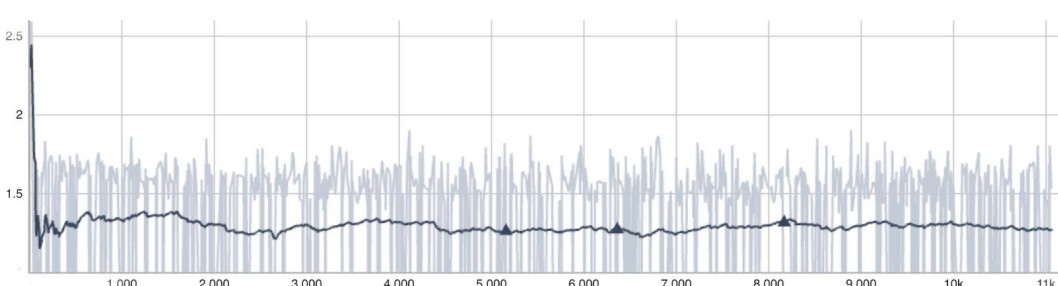

Figure 6: **Loss Curve of APEX SFT Training.**

# G VISUALIZATIONS PART I

This section provides additional qualitative results to complement the quantitative analysis in the main paper.

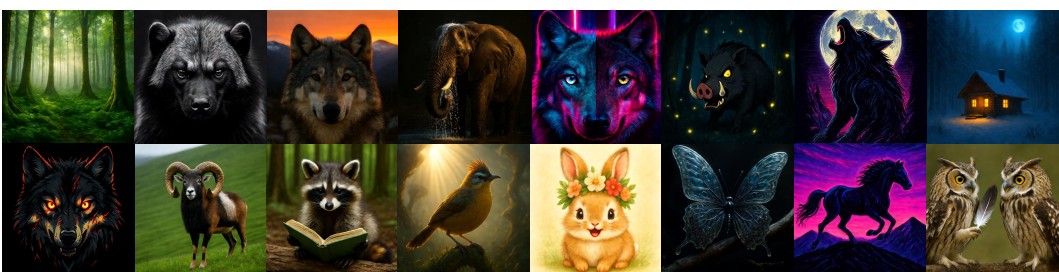

Figure 7: **Qualitative Comparison of 512x512 in APEX 20B LoRA for NFE=1.**

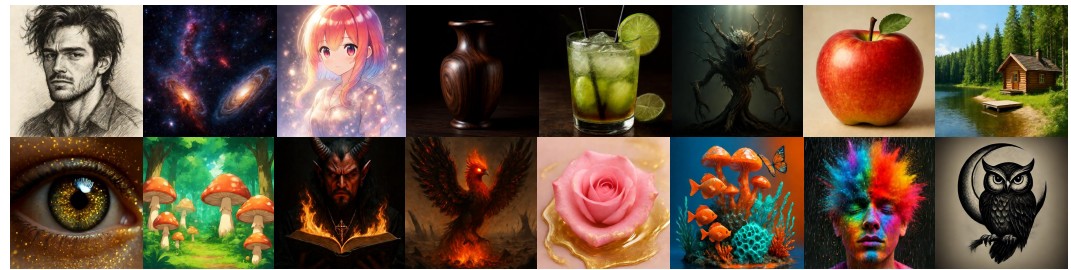

Figure 8: **Qualitative Comparison of 512x512 in APEX 20B LoRA for NFE=1.**

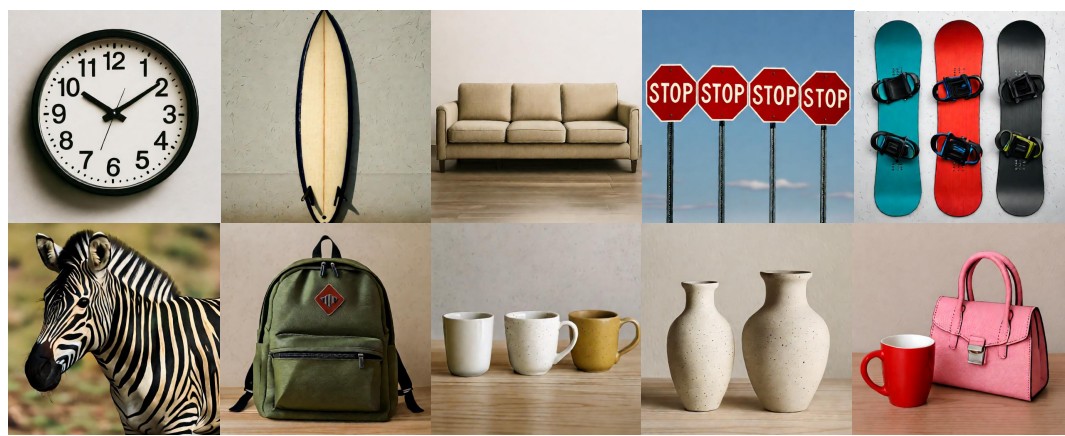

Figure 9: **Qualitative Comparison of 512x512 in APEX 20B LoRA for NFE=1.**

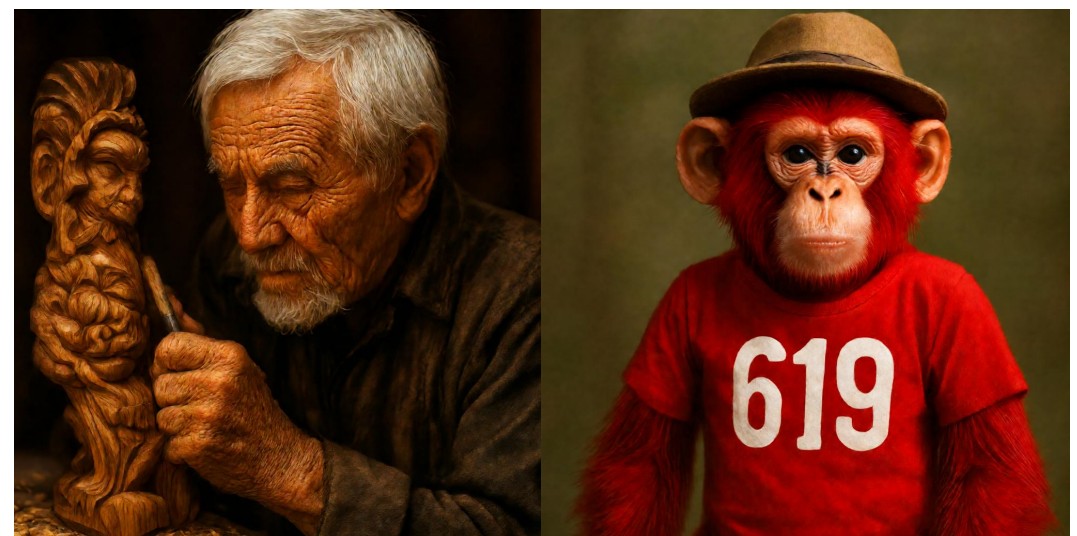

Figure 10: **Qualitative Comparison of 1024x1024 in APEX 20B LoRA for NFE=2.**

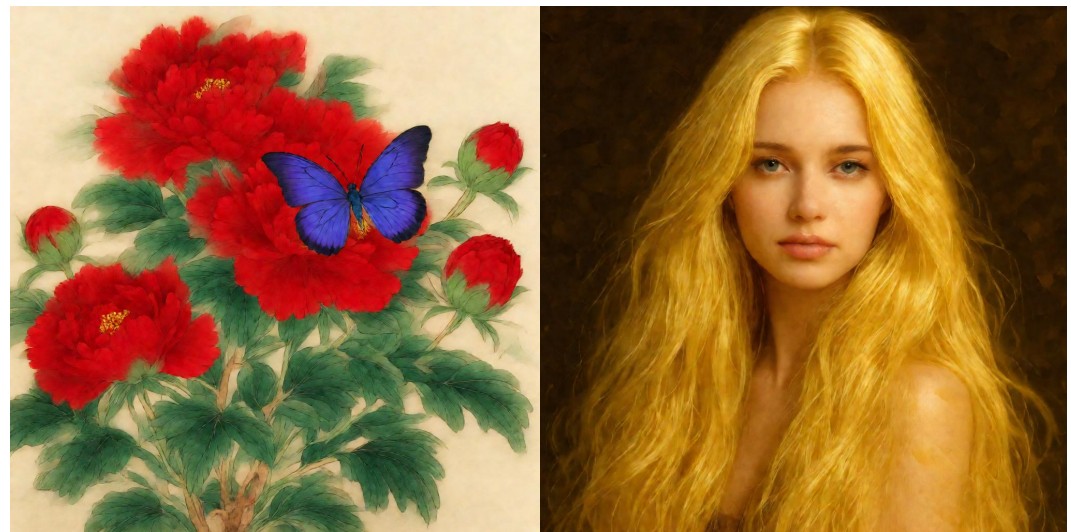

Figure 11: **Qualitative Comparison of 1024x1024 in APEX 20B LoRA for NFE=2.**

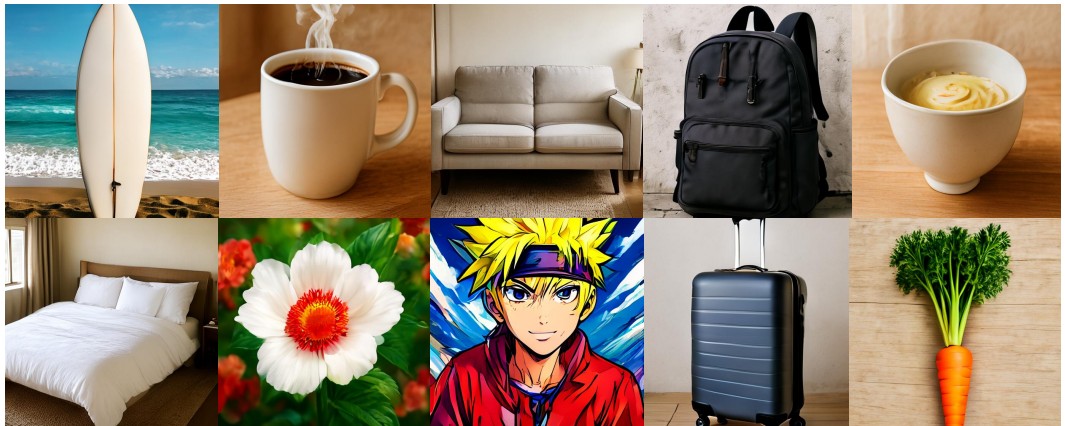

Figure 12: **Qualitative Comparison of 512x512 in APEX 0.6B LoRA for NFE=1.**

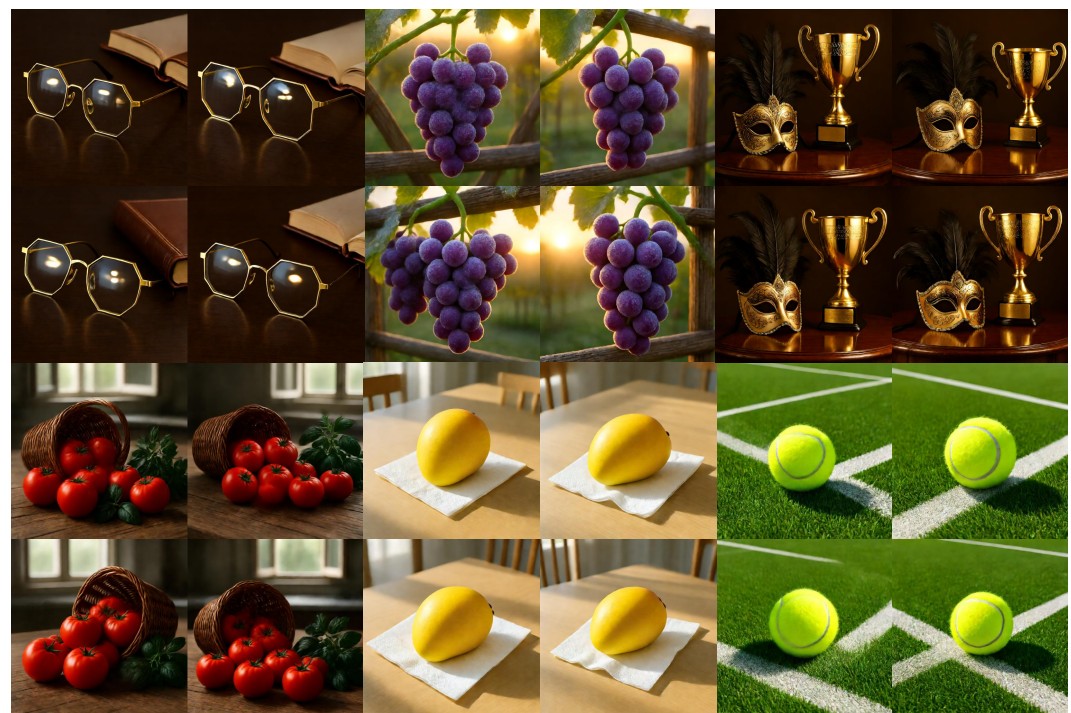

Figure 13: **Qualitative Comparison of 512x512 in APEX 20B LoRA for NFE=1.**

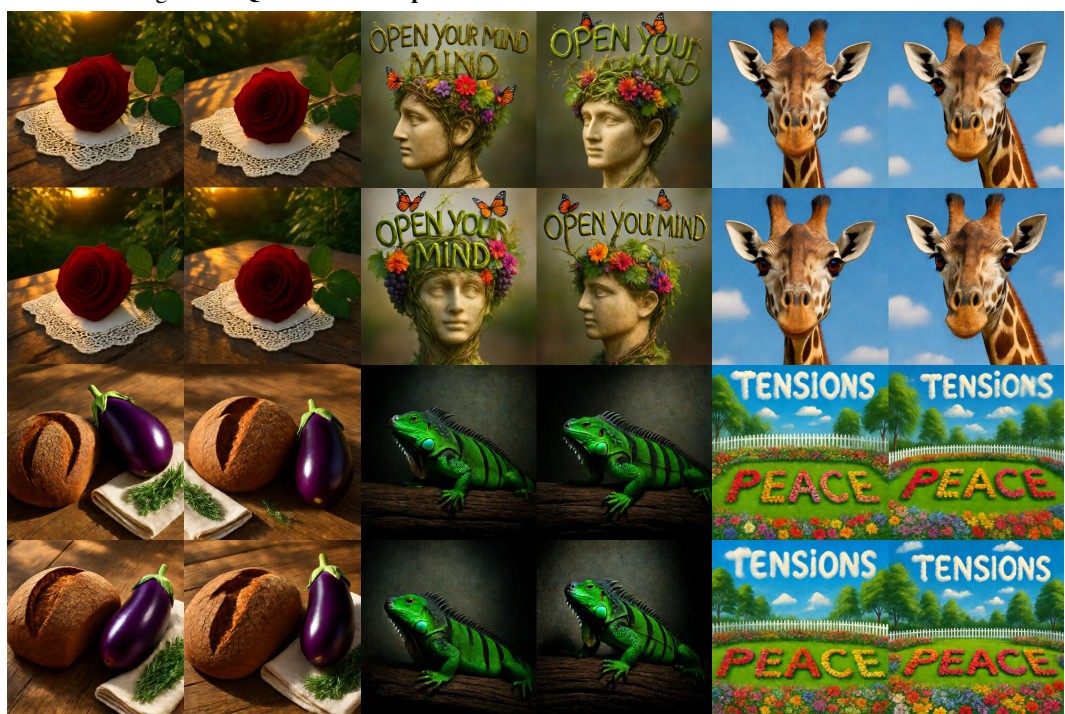

Figure 14: **Qualitative Comparison of 512x512 in APEX 20B LoRA for NFE=1.**

# H    VISUALIZATIONS PART II

# I    VISUALIZATIONS PART III

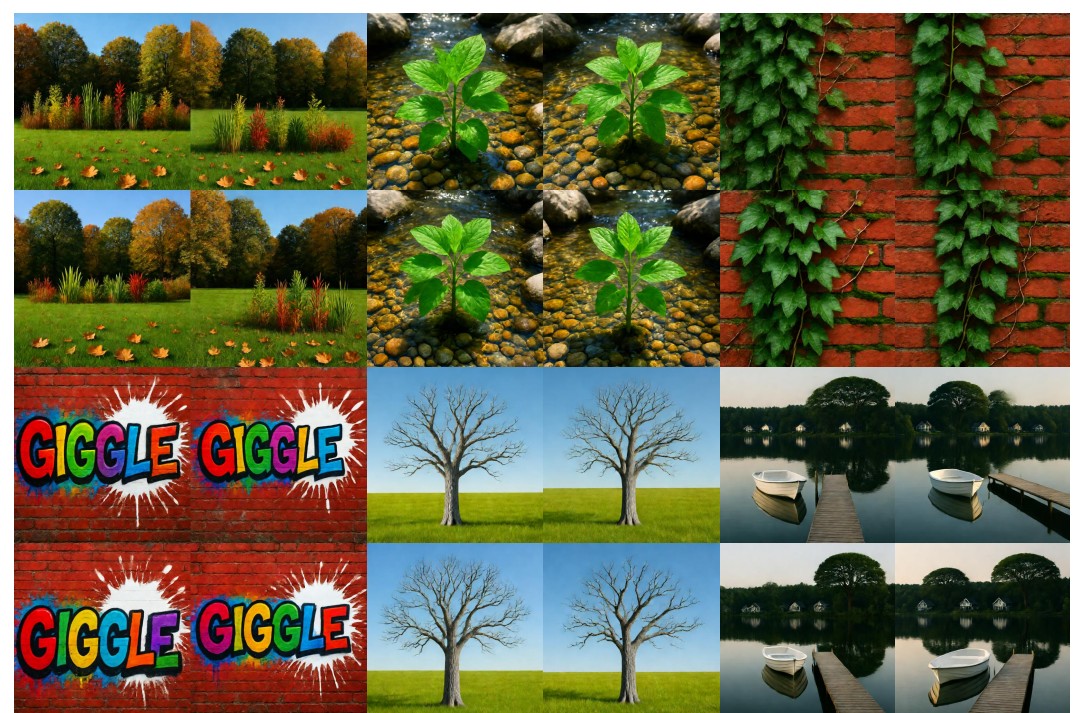

Figure 15: **Qualitative Comparison of 512x512 in APEX 20B LoRA for NFE=1.**

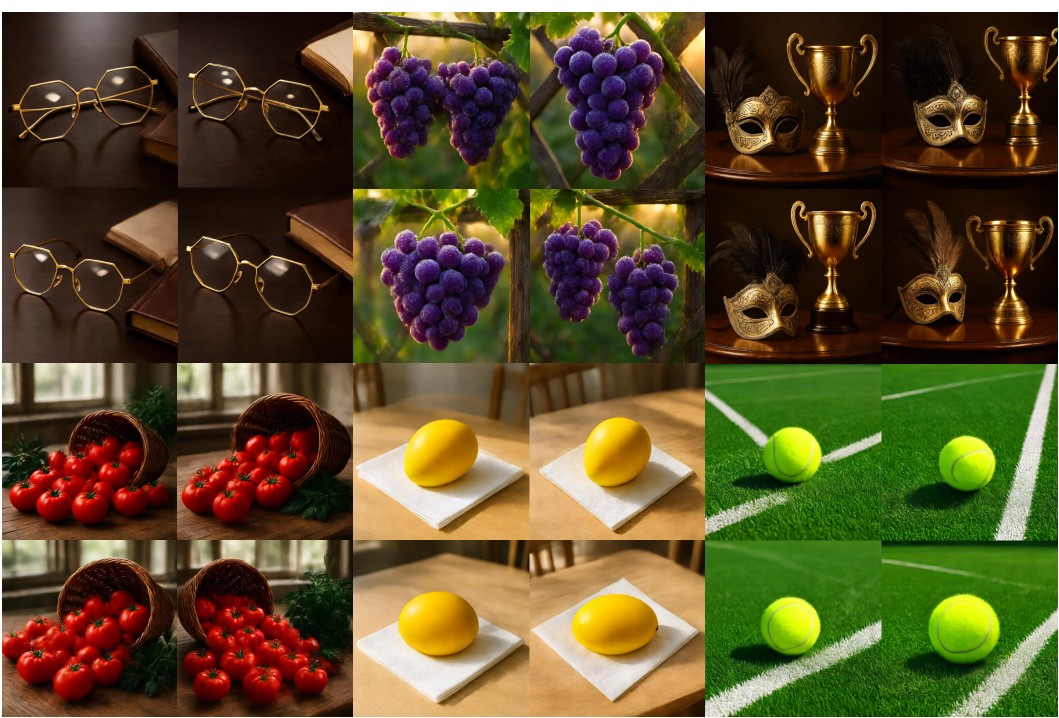

Figure 16: **Qualitative Comparison of 512x512 in APEX 20B Full Parameter Tuning for NFE=1.**

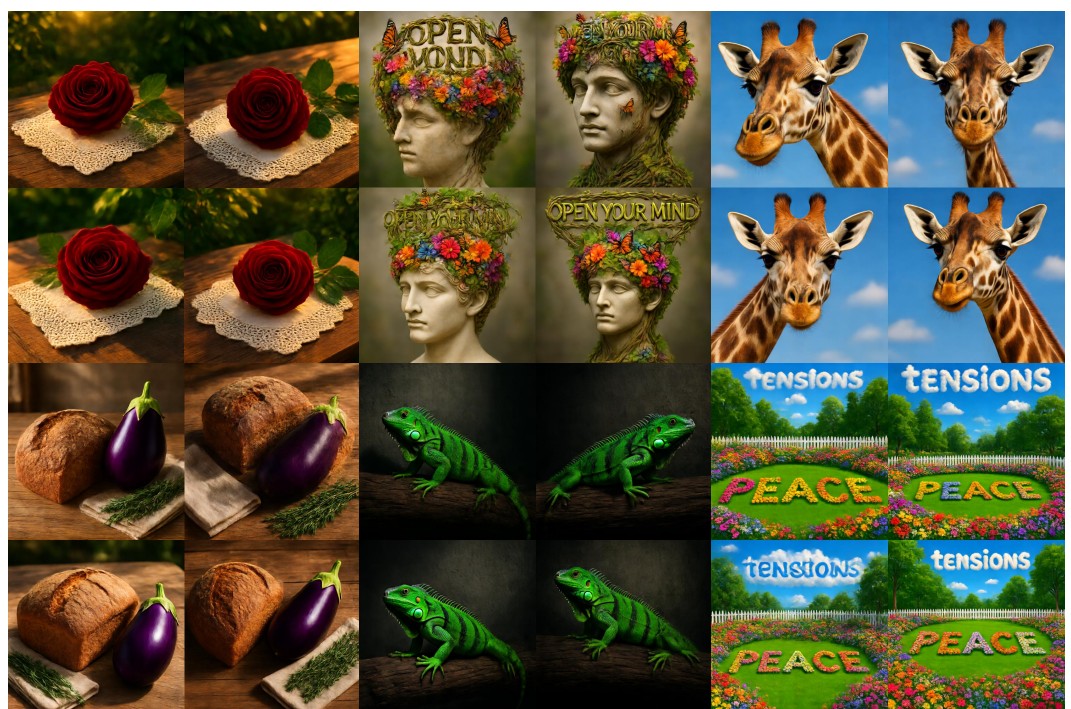

Figure 17: Qualitative Comparison of 512x512 in APEX 20B Full Parameter Tuning for NFE=1.

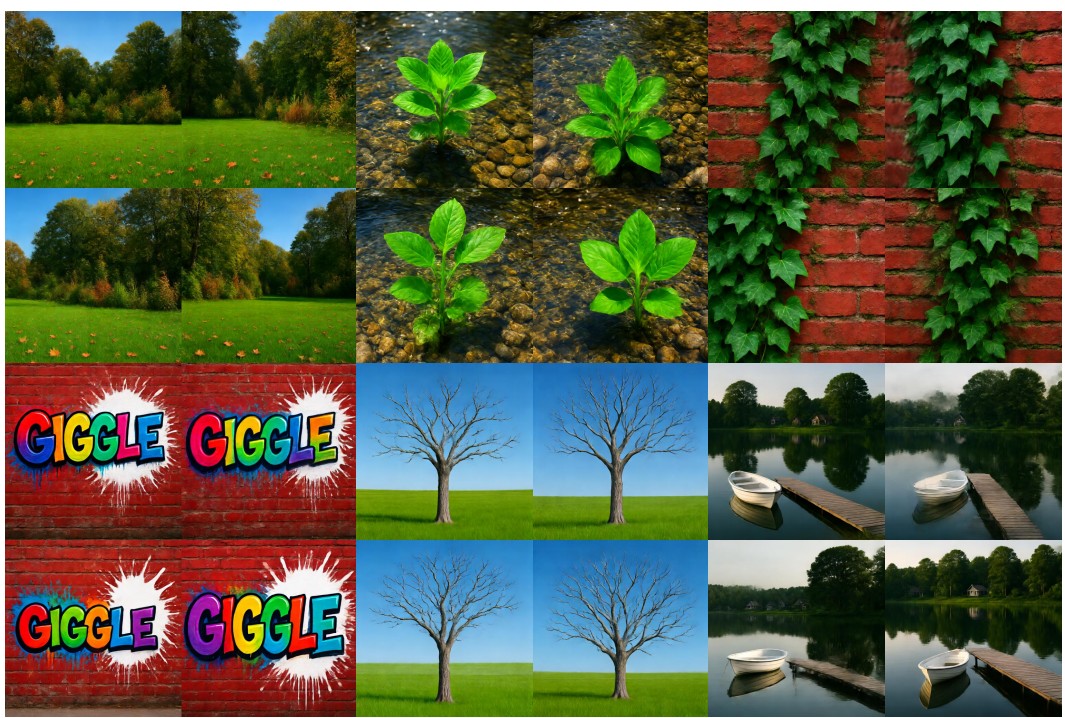

Figure 18: Qualitative Comparison of 512x512 in APEX 20B Full Parameter Tuning for NFE=1.

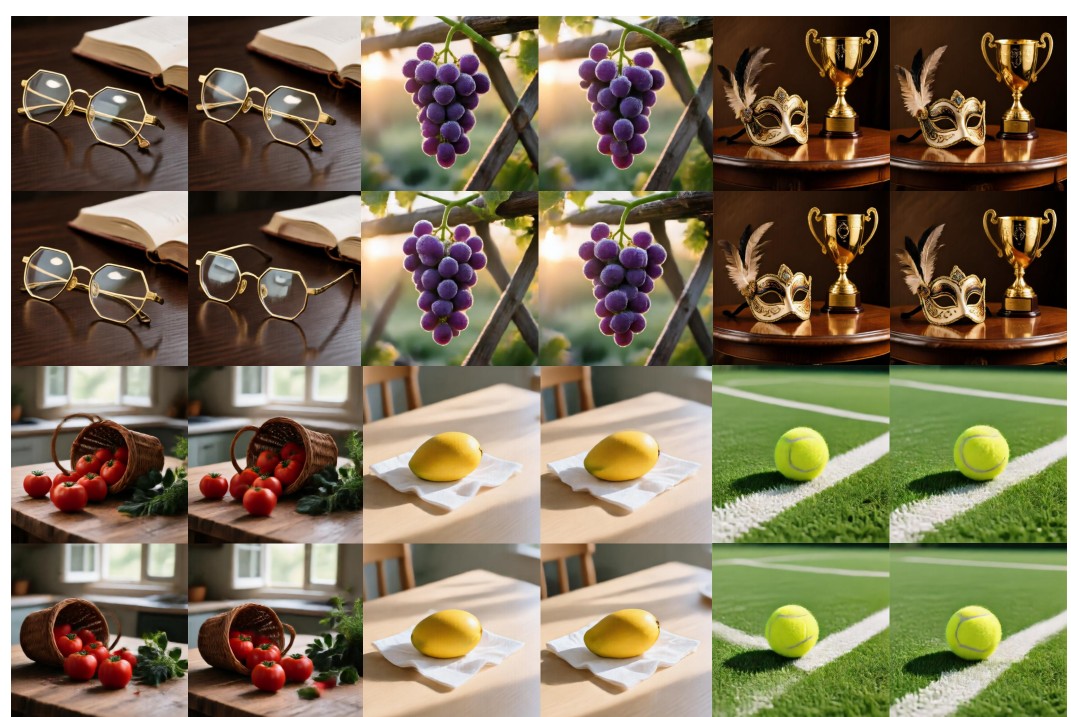

Figure 19: Qualitative Comparison of 512x512 in Qwen-Image Lightning LoRA for NFE=1.

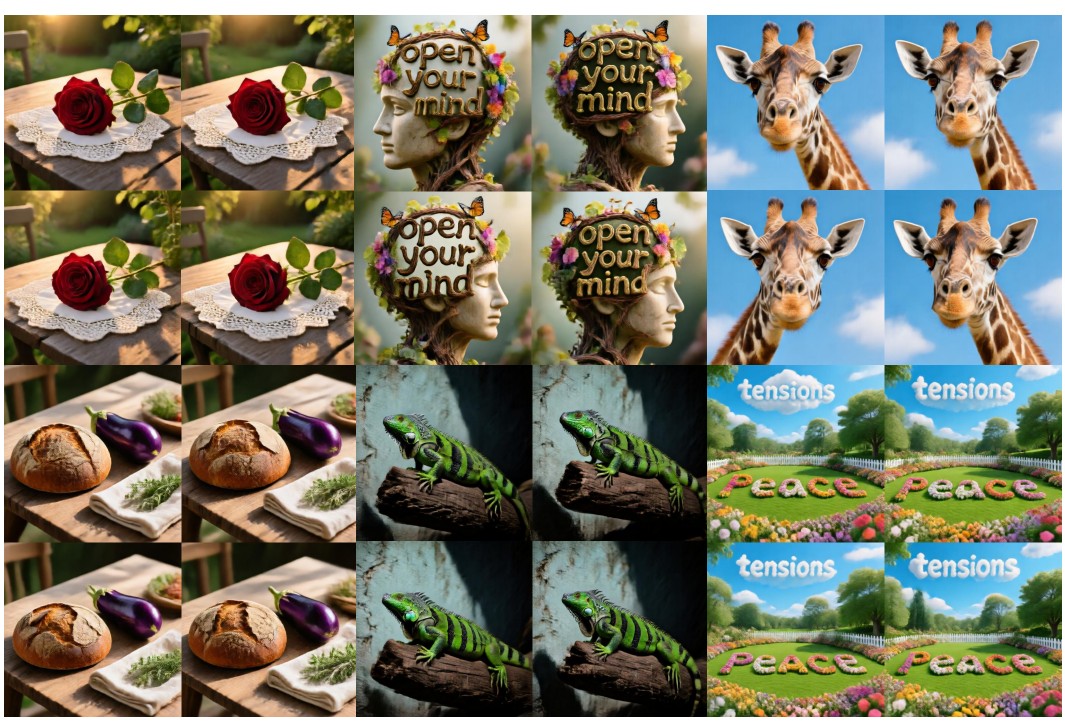

Figure 20: Qualitative Comparison of 512x512 in Qwen-Image Lightning LoRA for NFE=1.

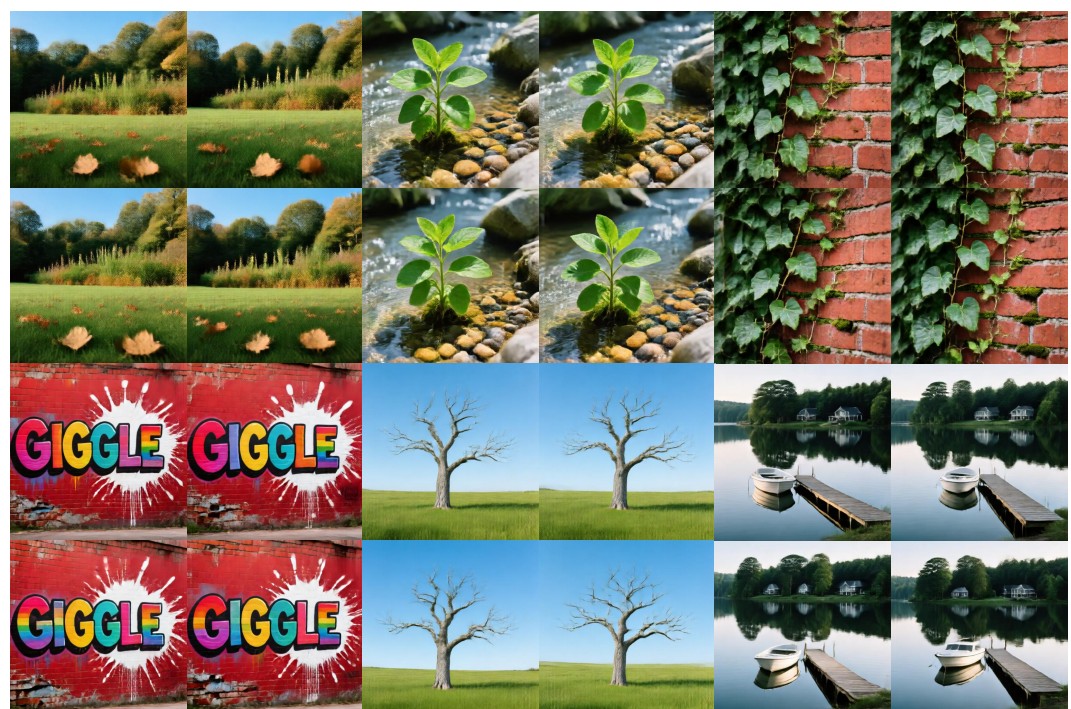

Figure 21: **Qualitative Comparison of 512x512 in Qwen-Image Lightning LoRA for NFE=1.**

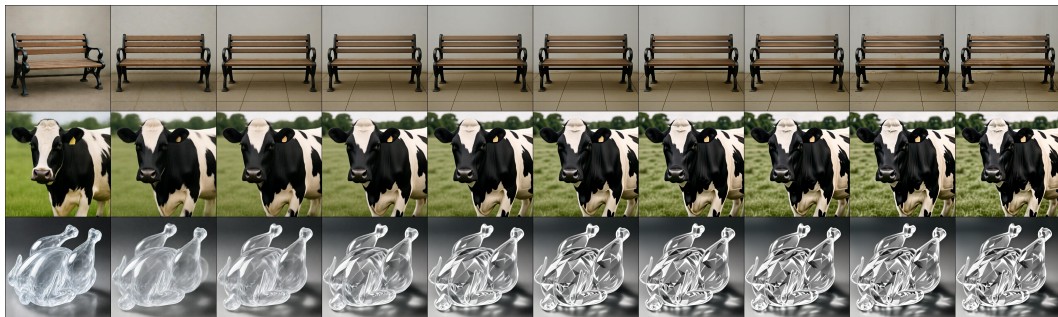

Figure 22: **Qualitative Comparison of 512x512 in 20B Full Parameter Tuning of APEX methods and Synthetic dataset from NFE=1 to NFE=20.**

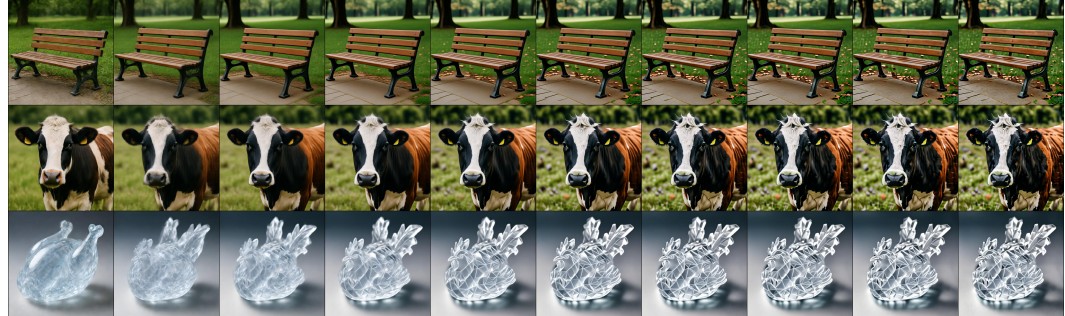

Figure 23: **Qualitative Comparison of 512x512 in 20B Full Parameter Tuning of APEX methods and BLIP-3o dataset from NFE=1 to NFE=20.**

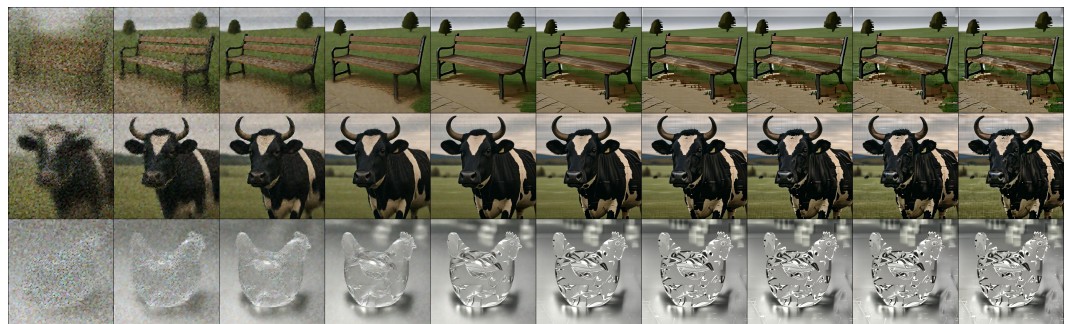

Figure 24: **Qualitative Comparison of 512x512 in 20B Full Parameter Tuning of sCM methods and BLIP-3o dataset from NFE=1 to NFE=20.**

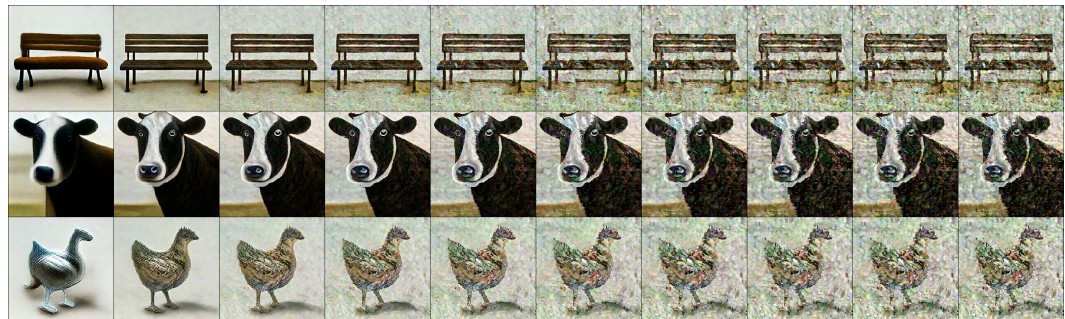

Figure 25: **Qualitative Comparison of 512x512 in 20B Full Parameter Tuning of CTM methods and BLIP-3o dataset from NFE=1 to NFE=20.**

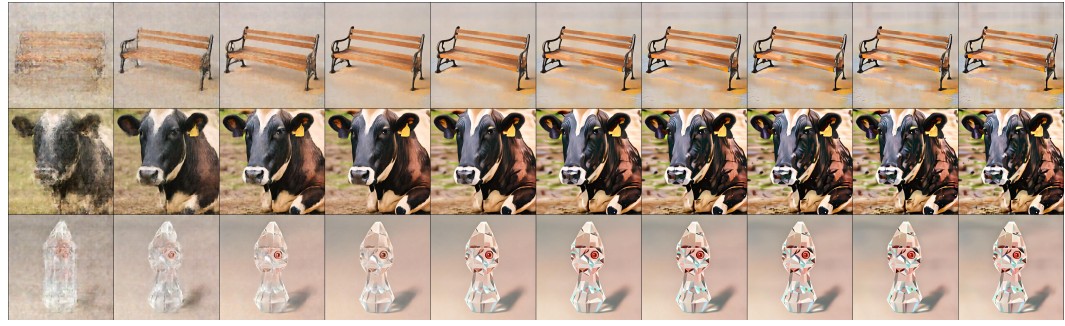

Figure 26: **Qualitative Comparison of 512x512 in 20B Full Parameter Tuning of MeanFlow methods and BLIP-3o dataset from NFE=1 to NFE=20.**

