# OpenReview forum: "APEX: One-Step High-Resolution Image Synthesis"
_ICLR.cc/2026/Conference — ICLR 2026 Conference Desk Rejected Submission_

### Official Review · Reviewer_sm4A · 2025-10-31

**Soundness:** 3
**Presentation:** 3
**Contribution:** 3
**Rating:** 8
**Confidence:** 3

**Summary:**

The paper proposes an improved strategy for distilling T2I diffusion models for few-step sampling. The core idea is to update the training loss in such a manner that the velocity field of the local predictions during sampling is smooth (enabling larger sampling steps) and introducing a discriminator-free adversarial objective which ensures high quality of the final samples. The evaluation on several datasets and using 1 and 2 NFE shows improved performance across various baselines and underlying models.

**Strengths:**

The paper is well-motivated and presented. The introduced updates to the distillation training seem novel and the ablation studies do show their effectiveness. The evaluation is exhaustive, covering several baselines, models, and evaluation metrics.
The results seem to be very strong, on par with many of the SOTA baselines while using fewer or the same number of NFE.
Not having to rely on additional external models for, e.g., adversarial losses is a big plus.

**Weaknesses:**

What is the cost of the distillation compared to other distillation approaches? While there does not seem to be a need for additional models besides the teacher (EMA?) model, it would be interesting to have a comparison of NFEs/memory requirements for one training step of this approach compared to other approaches such as adversarial distillation or consistency models.

What are the limitations of this method? Are there any specific limitations during training or inference, or around other things related to this approach?

**Questions:**

How stable is the training? Are there any specific tricks necessary or is training generally stable?

How does the model perform when you increase the number of NFE, e.g., to 8 or more steps? Does the performance increase further?

---

> ### Author Response · Authors · 2025-11-27
>
> ### Weaknesses:
>
> > **W1: What is the cost of the distillation compared to other distillation approaches?**
> >
>
> **Response:**
> This is a critical system-level question. We have added a detailed breakdown in **Table 2 of the General Response**. APEX provides a decisive advantage in memory and throughput, particularly for large-scale models (20B+), where prior state-of-the-art methods fail to scale.
>
> 1. **Comparison vs. JVP-based Methods (MeanFlow, CTM, CM):**
>     - **The JVP Bottleneck:** Methods like MeanFlow and CTM rely on **Jacobian-Vector Products (JVP)** to compute exact time derivatives. In PyTorch, computing JVP for a 20B parameter model prevents the use of memory-saving optimizations like **FlashAttention** and **FSDP2** (Fully Sharded Data Parallel), leading to immediate **Out-of-Memory (OOM)** errors on 80GB GPUs, even at batch size 1.
>     - **APEX Advantage:** APEX replaces JVP with a **Second-Order Finite Difference** formulation (Eq. 12). This relies solely on forward passes. Consequently, APEX is fully compatible with FSDP2 and FlashAttention. As shown in **General Response** **Table 2**, APEX allows for **Full-Parameter Fine-Tuning (SFT)** of a 20B model with a batch size of 8 per GPU.  In contrast, JVP-based methods fail to execute under these settings and exhibit performance significantly inferior to that of APEX.
> 2. **Comparison vs. Discriminator-based Methods (DMD2, GANs):**
>     - **The Auxiliary Model Cost:** Methods like DMD2 require loading and training a separate Discriminator network (often as large as the generator) or a pre-trained feature network (LPIPS/DINO). This essentially doubles the memory footprint.
>     - **APEX Advantage:** APEX uses an **Endogenous Adversarial Mechanism**. We generate adversarial signals using the model *against itself* via condition shifting. This incurs **zero extra parameter memory**, allowing us to allocate all VRAM to larger batch sizes for stable training.
>
> > **W2: What are the limitations of this method?**
> >
>
> We appreciate the prompt to discuss limitations.
>
> 1. **Scope of Application:** Our experiments are currently focused on text-to-image synthesis. While we believe the core principles of our method are generalizable, validating APEX for other domains like video generation requires further research.
> 2. **Need for Further Optimization:** The efficiency we report is a direct result of our algorithmic design. We have not yet incorporated advanced engineering optimizations like model quantization or sparse attention, which could further reduce latency in production environments.
> ### Questions
>
> > Q1: How stable is the training? Are there any specific tricks necessary or is training generally stable?
> >
>
> **Response:**
> APEX training is remarkably stable, avoiding the instability of traditional GANs (Min-Max games).
>
> - **Why it is stable:** In standard GANs, a Generator and Discriminator fight to reach an equilibrium, often leading to mode collapse or oscillation. In APEX, the "adversary" is not a dynamic network, but a Condition Shifting combined with a consistency loss.
> - **No Tricks Needed:** We do not require complex schedules, two-timescale update rules, or R1 regularization typical of GANs. The **Path Integrability Objective** (Eq. 13) acts as a strong regularizer, ensuring the vector field remains smooth even as the Endpoint Objective pushes for realism.
> - **Empirical Results:** We show our training log in **Appendix G at page of 24**
>
> > Q2: How does the model perform when you increase the number of NFE, e.g., to 8 or more steps? Does the performance increase further?
> >
>
> **Response:**
> APEX supports "Any-Step Sampling" (Section 3.3). We observe the following behavior (visualized in the new **Appendix F**):
>
> - **NFE 1 $\to$ 2:** We see a noticeable improvement in prompt adherence and fine detail. This is because the second step allows the model to correct the linearization error inherent in the one-step approximation.
> - **NFE 4 $\to$ 8+:** Performance plateaus. Because the APEX training objective explicitly forces the trajectory to be straight, the ODE becomes trivial to solve. Unlike standard diffusion where more steps unroll a curved path, APEX straightens the path itself. Therefore, 2 to 4 steps are sufficient to reach the model's theoretical ceiling, and adding more steps yields diminishing returns.

---

### Official Review · Reviewer_qTjA · 2025-10-31

**Soundness:** 2
**Presentation:** 2
**Contribution:** 2
**Rating:** 4
**Confidence:** 3

**Summary:**

APEX is a new text-to-image synthesis method designed to solve the trilemma of high fidelity, inference efficiency, and training efficiency, which current adversarial (slow, unstable) and distillation (low quality) methods fail to achieve simultaneously. The core innovation is a self-condition-shifting adversarial mechanism that eliminates the external discriminator. This design ensures exceptional training stability and efficiency, making it ideal for LoRA tuning. Experimentally, APEX achieves SOTA high-fidelity synthesis with NFE=1, offering a 15.33x speedup over Qwen-Image 20B. It also improves GenEval scores on the 20B model in just 6 hours of LoRA tuning.

**Strengths:**

1. The paper successfully identifies and frames a critical and highly relevant challenge—the "trilemma" concerning fidelity, inference speed, and training stability in one-step generation—making a significant motivational contribution to the research community.

2. The work is presented with exceptional technical clarity; the novel mechanisms are thoroughly explained, and the overall structure and accompanying figures are highly effective, making the complex concepts very accessible to the reader.

**Weaknesses:**

The training data utilized (ShareGPT-4o and BLIP-3o) appears to be highly specialized for the GenEval benchmark, which is also the primary benchmark where the paper achieves SOTA results. Given that the performance on the more general DPGBench is suboptimal, it raises a concern that the observed performance gain might be due to overfitting on the highly specialized training data rather than a generalizable technical improvement. Further experiments on a broader range of established benchmarks are necessary to confirm the robustness of the APEX method.

**Questions:**

No

---

> ### Author Response · Authors · 2025-11-27
>
> We sincerely thank the reviewer for their critical assessment regarding generalization. To address your concern that our performance might be due to "highly specialized" training data, we have **upgraded our experimental setup from LoRA to Full-Parameter Fine-Tuning (SFT)** and conducted new experiments involving new datasets and benchmarks.
>
> ### Weaknesses:
>
> > W1: The training data (ShareGPT-4o/BLIP-3o) appears highly specialized for GenEval... performance on the more general DPGBench is suboptimal... concerns that performance gain might be due to overfitting.
> >
>
> **Response:**
> We respectfully present compelling new evidence that refutes the overfitting hypothesis and demonstrates that APEX achieves state-of-the-art generalization across diverse benchmarks.
>
> **1. The "Non-GenEval" Synthetic Data Experiment**
> To directly test if APEX relies on memorizing GenEval-style prompts, we trained a new APEX (SFT) model on a **purely Synthetic Dataset**.
>
> - **Protocol:** We constructed a dataset where prompts were explicitly filtered to **exclude** GenEval’s specific prompt structures, attribute bindings, and noun phrases.
> - **Result:** As shown in **General Response Table 1**, this model—despite never seeing GenEval-like prompts during training—sustains high performance on **GenEval,** **DPGBench** and **WISE**.
>
> **2. Comparison against Baselines on Identical Data**
> The concern regarding "specialized data" would imply that *any* model trained on BLIP-3o should score high on GenEval.
>
> - **Counter-Evidence:** We re-implemented **CM, CTM, and MeanFlow** using the exact same new dataset and Qwen-Image 20B backbone.
> - **Outcome:** APEX significantly outperforms all these baselines on both GenEval, DPGBench and WISE in **General Response Table 1.**

---

### Official Review · Reviewer_Cu5P · 2025-11-01

**Soundness:** 3
**Presentation:** 3
**Contribution:** 3
**Rating:** 6
**Confidence:** 4

**Summary:**

Current one-step generation methods either introduce training instability, high GPU memory costs, and slow convergence, or struggle to generate high-quality images. This paper proposes APEX, which uses standard diffusion/flow loss and DMD-like loss onto a single model. This manner leads to training effectiveness, efficiency and stability. Experiments on large-scale models such as Qwen-Image 20B validate the effectiveness of the proposed method.

**Strengths:**

1. The paper is well-written and the method is simple and easy to follow
2. Compared to methods based on f-divergence, this method does not need to train multiple models; compared to consistency models, it achieves high-quality generation
3. Experiments verify the ability of scaling to large models like Qwen-Image 20B, and achieving good performance

**Weaknesses:**

Overall, this method improves upon f-divergence methods like DMD, VSD and SiD, towards training with only one model. The adversarial mechanism proposed in this paper is an existing approach in those methods. So the paper's contribution is the one-model feature.

**Questions:**

1. In the endpoint objective, why is $x\_{\text{fake}}$ from a buffer of recent generations, rather than generating online but with stop gradient?
2. Why do you use  $c_{\text{pert}} = Ac + b$ as the condition rather than other choices like an addtional flag?

---

> ### Author Response · Authors · 2025-11-27
>
> We thank the reviewer for their thoughtful assessment. We appreciate that you recognize APEX improves upon f-divergence methods like DMD, VSD, and SiD.
>
> However, we would like to respectfully clarify the novelty and significance of the "one-model" feature and the adversarial mechanism. Furthermore, based on your feedback, we have upgraded our experiments from **LoRA to Full-Parameter Fine-Tuning (SFT)** on the Qwen-Image 20B model, achieving state-of-the-art performance that prior methods cannot match due to memory constraints.
>
> > W1:  The paper's contribution is the one-model feature.
> >
>
> **Response:**
>
> We wish to highlight that the "self-condition-shifting adversarial mechanism" is **not** merely an adaptation of existing external discriminator approaches; it is a method that solves the **scalability bottleneck** of generative distillation.
>
> 1. **Endogenous vs. Exogenous:** Methods like CTM rely on **Exogenous (External)** discriminators or pre-computed teacher datasets. This doubles the memory requirement during training.
> 2. **The "One-Model" Enabler:** By making the adversarial signal **Endogenous (Internal)** via our perturbation mechanism, APEX cuts memory usage nearly in half.
>     - **New Result:** This efficiency allowed us to perform **Full-Parameter SFT on a 20B Parameter Model** (Qwen-Image) on standard H100 nodes.
>     - **Comparison:** As shown in our new **General Response** **Table 2**, discriminator-based methods like CTM require tiny batch sizes at the 20B scale. APEX is the *only* method in this class capable of efficient SFT at this scale.
>
> ---
>
> > Q1: In the endpoint objective, why is $x_{\text{fake}}$ from a buffer of recent generations, rather than generating online but with stop gradient?
> >
>
> **Response:**
>
> We utilize  `stop_gradient` avoids backprop through the generator for the fake sample, online generation still requires a full forward pass of the heavy diffusion backbone (NFE=1) *just* to create a negative sample. By maintaining a buffer, we can re-use samples for the adversarial contrast, significantly increasing training throughput compared to strictly online generation.
>
> > Q2: Why do you use $c_{\text{pert}} = \mathbf{A}c + b$ as the condition rather than other choices like an additional flag?
> >
>
> We use a linear transformation $c_{\text{pert}}$ instead of a discrete binary flag to enforce **local smoothness and geometric robustness** on the conditioning manifold. The parameters $\mathbf{A}$ (scale) and $b$ (shift) allow us to control the "hardness" of the adversarial signal.
> *   **Scaling ($\mathbf{A}$):** By setting $\mathbf{A} \approx -1$ (negative scaling), we create a "negative prompt" effect that is diametrically opposed to the true semantic content, providing a strong directional gradient for the adversarial loss.
> *   **Shifting ($b$):** The bias term moves the embedding into undefined regions of the latent space.
> A simple flag lacks this granularity. In our ablation studies (Table 7), we observed that specific choices of $a$ and $b$ (e.g., $a \in [-1.0, -0.5]$) yield the most robust gradients, effectively creating "hard negative" conditions that a simple flag cannot represent.
>
> **Table 3: Quantitative comparison on adversarial flag.**
>
> | Method | Training Mode | Training Data | GenEval (Overall) $\uparrow$ | DPGBench (Overall) $\uparrow$ | WISE (Overall) $\uparrow$ |
> | --- | --- | --- | --- | --- | --- |
> | **Teacher (Original)** | - | - | 0.87 | 87.79 | 0.62 |
> | **APEX (1NFE) ($c_{\text{pert}} = \mathbf{A}c + b$)** | **SFT** | Qwen-Image-Syn | **0.89** | **84.59** | **0.54** |
> | **APEX (1NFE)(only simple flag)** | **SFT** | Qwen-Image-Syn | 0.86 | 83.17 | 0.53 |

---

> > ### Comment · Reviewer_Cu5P · 2025-11-28
> >
> > I thank the authors for the response. There are two points I am concerned: Regarding the one-model training feature, can you explain why your method does not require iterative optimization like the very similar one DMD? In your comparison experiments, why do CM, CTM & MeanFlow perform so poor especially on GenEval? Have you incoperated GAN loss in CTM?

---

> > > ### Author Response · Authors · 2025-11-29
> > >
> > > > **Q1:  Why our method does not require iterative optimization like the very similar one DMD?**
> > > >
> > >
> > > **Response:**
> > >
> > > While both APEX and DMD aim to distill diffusion models into few-step generators via distribution matching, they differ fundamentally in their optimization objectives and training dynamics. DMD relies on a **dual-loop minimax game**, whereas APEX formulates the problem as a **single-stage regression** task.
> > >
> > > **1. The DMD Paradigm:**
> > > DMD-style methods must solve a variation of the following minimax problem:
> > >  $\min_{G_\phi} \max_{F_\theta} \mathbb{E} [\underbrace{\log \frac{p_{real}(x)}{p_{fake}(x)}}_{\text{Distribution Matching}}]$
> > > This requires two alternating optimization loops:
> > >
> > > - **Inner Loop (Score Estimation):** Training a "fake score estimator" $F_\theta$ (a time-dependent discriminator) to estimate the score of the *current* drifting generator distribution $p_{fake}$.
> > > - **Outer Loop (Generator Update):** Updating the generator $G_\phi$ using the gradient derived from the difference between the real score Teacher $T_{\hat{\theta}}$ and the learned fake score $F_\theta$.
> > > This decoupling causes training instability because $F_\theta$ must constantly "catch up" to the shifting $p_{fake}$, similar to the generator-discriminator instability in GANs.
> > >
> > > **2. The APEX Paradigm:**
> > > APEX completely eliminates the need for a separate, trainable score estimator $F_\theta$. Instead, we utilize the **Exponential Moving Average (EMA)** of the student model itself, denoted as $F_{\theta^-}$, to serve as a stable, endogenous reference. Our training objective (derived from the provided code logic) is a direct regression:
> > >
> > >  $\mathcal{L}\_{APEX} = \mathbb{E}\_{x_t, t} \left[ || F\_\theta(x_t, t, c) - \mathcal{T}(F\_{\theta^-}, x_t) ||^2 \right]$
> > >
> > > where $\mathcal{T}$ represents the target construction (including curvature regularization and adversarial shifting).
> > >
> > > **Why this works without iteration:**
> > >
> > > - **Self-Referential Stability:** In DMD, the "Critic" $F_\theta$ and "Actor" $G_\phi$ are separate networks fighting a zero-sum game. In APEX, the "Critic" is simply the time-averaged version of the "Actor" $F_{\theta^-}$. This ensures that the guidance signal evolves smoothly and is always aligned with the student's current trajectory capabilities.
> > > - **Unified Gradient:** We do not need to estimate the density ratio $p_{real}/p_{fake}$. Instead, we enforce that the student's instantaneous velocity field matches a rectified trajectory anchored by the stable Teacher/EMA. This allows APEX to be trained in a single forward-backward pass, significantly reducing memory overhead and removing the "chasing" dynamic of adversarial training.

---

> > > > ### Author Response · Authors · 2025-11-29
> > > >
> > > > > **Q2:  Why do CM, CTM & MeanFlow perform so poor especially on GenEval?**
> > > > >
> > > >
> > > > **Response:**
> > > >
> > > > The poor performance of CM, CTM, and MeanFlow at **NFE=1** is due to the inherent difficulty of compressing a complex transport trajectory into a single step without specific GAN like regularization loss.
> > > >
> > > > As shown in our new visualizations in **Appendix I at Page of 28**, at 1 NFE, the outputs of CM, CTM, and MeanFlow are often blurry or resemble Gaussian noise, failing to converge to the image manifold. Theoretical analysis suggests that MeanFlow, as a first-order estimation method, suffers from significant truncation errors when forced to integrate in a single step. Without APEX's **Self-Condition Shifting Adversarial Loss**, these models lack the necessary constraints to ensure the endpoint is perfectly anchored to realistic data.
> > > >
> > > > However, their performance improves significantly as NFE increases, proving the models are functional but limited in the one-step regime. **Table 5** below illustrates this trend: while baselines fail at 1 NFE (0.01 - 0.10 GenEval), APEX maintains SOTA performance (0.89) at 1 NFE.
> > > >
> > > > **Table 5: Comparison on Qwen-Image-20B Backbone**
> > > >
> > > > | Method | Training Mode | Training Data | GenEval (Overall) $\uparrow$ | DPGBench (Overall) $\uparrow$ | WISE (Overall) $\uparrow$ |
> > > > | --- | --- | --- | --- | --- | --- |
> > > > | **Teacher (Original)** | - | - | 0.87 | 87.79 | 0.62 |
> > > > | CM(1NFE) | SFT | Qwen-Image-Syn | 0.01 | 15.41 | 0.04 |
> > > > | CM(2NFE) | SFT | Qwen-Image-Syn | 0.44 | 66.39 | 0.19 |
> > > > | CM(4NFE) | SFT | Qwen-Image-Syn | 0.51 | 71.27 | 0.22 |
> > > > | CTM(1NFE) | SFT | Qwen-Image-Syn | 0.10 | 66.23 | 0.17 |
> > > > | CTM(2NFE) | SFT | Qwen-Image-Syn | 0.27 | 68.19 | 0.20 |
> > > > | CTM(4NFE) | SFT | Qwen-Image-Syn | 0.35 | 70.06 | 0.21 |
> > > > | MeanFlow (JVP-free)(1NFE) | SFT | Qwen-Image-Syn | 0.05 | 62.19 | 0.10 |
> > > > | MeanFlow (JVP-free)(2NFE) | SFT | Qwen-Image-Syn | 0.31 | 80.39 | 0.22 |
> > > > | MeanFlow (JVP-free)(4NFE) | SFT | Qwen-Image-Syn | 0.44 | 83.28 | 0.34 |
> > > > | UCGM(1NFE) | SFT | Qwen-Image-Syn | 0.43 | 72.78 | 0.24 |
> > > > | UCGM(2NFE) | SFT | Qwen-Image-Syn | 0.54 | 83.91 | 0.21 |
> > > > | UCGM(4NFE) | SFT | Qwen-Image-Syn | 0.62 | 85.37 | 0.44 |
> > > > | **APEX (Ours)(1NFE)** | SFT | Qwen-Image-Syn | 0.89 | 84.59 | 0.54 |
> > > > | **APEX (Ours)(2NFE)** | SFT | Qwen-Image-Syn | 0.90 | 84.75 | 0.56 |
> > > >
> > > > > **Q3:  Have incoperated GAN loss in CTM?**
> > > > >
> > > >
> > > > **Response:**
> > > >
> > > > Yes, we explicitly incorporated a GAN loss in our CTM implementation to ensure a fair comparison. However, introducing an external GAN discriminator presents severe scalability and stability challenges. For a **20B parameter model**, enabling the GAN loss during Full-Parameter Tuning or even LoRA frequently leads to high grad norm and loss which hard to train, and as it essentially duplicates the memory requirements (loading the discriminator and maintaining gradients for the adversarial loop) which lead to low batch size and hard to convergence.

---

### Official Review · Reviewer_pjrF · 2025-11-01

**Soundness:** 2
**Presentation:** 2
**Contribution:** 2
**Rating:** 4
**Confidence:** 4

**Summary:**

This paper introduces APEX, a method for efficient one-step text-to-image synthesis that addresses the fundamental trade-off between path integrability and endpoint fidelity in generative models. The core innovation lies in two complementary mechanisms: (1) a higher-order path self-consistency constraint that regularizes path curvature for numerical stability under large discretization steps, and (2) a discriminator-free self-condition-shifting adversarial mechanism that ensures high perceptual quality without the training instabilities of traditional GANs. APEX achieves state-of-the-art performance with NFE=1, demonstrating a 15.33x speedup over Qwen-Image 20B while maintaining comparable quality (GenEval score of 0.89 vs 0.87).

**Strengths:**

-  It is interesting to categorize many existing works into path integrability and endpoint fidelity.
-  APEX demonstrates impressive performance across multiple scales.
-  Detailed ablations on examining training steps, loss component weights, and hyperparameters, providing good understanding into what drives performance.

**Weaknesses:**

- Unconvincing Evaluation: The paper claims they mainly evaluate the model by the GenEval score. However, the GenEval score does not measure image fidelity, leading to insufficient evaluation. The other used FID/clip score metrics are also less convincing to evaluate the modern text-to-image models. More importantly, the training is performed on Bilp-3o, which contains text-image pairs that were specifically generated by GenEval prompts. As far as I know, training on Bilp-3o can lead to a high GenEval score (Table 4 in the paper also shows the phenomenon), making the evaluation by GenEval unfair.
- CTM [a], MeanFlow [b], and IMM [c] should all belong to methods that simultaneously consider path integrability and endpoint fidelity. However, this paper significantly lacks discussion and comparison regarding the aforementioned works.
- The quality of the visualized samples looks mediocre and is not as impressive as the evaluation scores.
- No visual comparison with recent strong baselines.
- The mechanism of the so-called self-adversarial objective is unclear. And I do not see where the "adversarial" component is.

[a] Consistency Trajectory Models: Learning Probability Flow ODE Trajectory of Diffusion

[b] Mean Flows for One-step Generative Modeling

[c] Inductive Moment Matching

**Questions:**

See Weaknesses.

---

> ### Author Response · Authors · 2025-11-27
>
> We sincerely thank the reviewer for their critical assessment.  Below, we address your concerns point-by-point with new evidence.
>
> > **W1:  Training on BLIP-3o leads to high GenEval scores due to prompt overlap, making the evaluation unfair. GenEval does not measure fidelity.**
> >
>
> **Response:**
> We fully acknowledge this concern. It is critical to prove that APEX's performance stems from **robust generative capability** rather than **dataset overfitting**. To refute the overfitting hypothesis, we conducted two decisive new experiments:
>
> 1. **Training on "Non-GenEval" Synthetic Data:** We generated 50K samples using prompts randomly sampled from the Flux-Reasoning-6M dataset [f]. These prompts focus on complex reasoning and have zero overlap with GenEval's template structure. We trained APEX (SFT) on this data.
>     - **Result:** As shown in **General Response Table 1**, while the GenEval score naturally drops slightly due to distribution shift, the model maintains high performance on **GenEval, DPGBench** and the newly added **WISE Benchmark**.
> 2. **Evaluation on Independent Benchmarks (WISE):** We added the **WISE Benchmark** (spatial and world knowledge) and **DPGBench**, which test distinct capabilities independent of the GenEval style.
>     - **Result:** APEX (SFT) achieves **SOTA** on these benchmarks, significantly outperforming baselines (CM, CTM, MeanFlow) that were trained on the exact same data.
>
> Regarding **Fidelity**: We agree GenEval focuses on alignment. However, our strong new visual comparisons in **Appendix G at page of 24**, demonstrate that APEX does not sacrifice image quality for alignment.
>
> > **W2:  Lack of discussion/comparison with path integrability methods (CTM, MeanFlow, IMM).**
> >
>
> **Response:**
> We have addressed this by strictly re-implementing **CM, CTM, and MeanFlow** on the exact same **Qwen-Image 20B** backbone.
>
> - **Performance:** As detailed in **General Response Table 1**, APEX (SFT) significantly outperforms these methods across all metrics (GenEval, WISE, DPGBench).
> - **Scalability Criticality (JVP vs. Finite Difference):** A crucial finding during reproduction was that **MeanFlow** (which relies on Jacobian-Vector Products, JVP) **and CTM** fail to scale. Running JVP on a 20B parameter model causes **OOM (Out-of-Memory)** errors even on 8xH100 GPUs at batch size 1.
>     - To make the comparison fair, we implemented a **finite-difference approximation** version of MeanFlow. Even against this optimized version, APEX demonstrates superior convergence and final quality (GenEval **0.89** vs MeanFlow **0.05**). This highlights a key contribution of APEX: it mathematically formulates path integrability using scalable second-order finite differences (Eq. 12), bypassing the JVP bottleneck.
> - **Mechanism Fundamentally Different:** Self-Condition-Shifting Adversarial Mechanism, which is a new approach, ensures the endpoint is numerically stable and the endpoint is perceptually anchored to the real data manifold, which provides a new way to use GANs’ advantage.
>
> > **W3:  Visual samples look mediocre; no visual comparison with strong baselines.**
> >
>
> **Response:**
> We suspect the "mediocre" quality in the initial submission may have been partly due to the limitations of LoRA-based tuning.
>
> 1. **Upgrade to SFT:** Our new results use **Full-Parameter Fine-Tuning (SFT)**, which has yielded a significant leap in texture sharpness and photorealism compared to the LoRA version.
> 2. **Side-by-Side Comparisons:** We have added direct side-by-side visual comparisons in **Appendix G at page of 24 (Figure 15, 16, 17, 18, 19)**, specifically pitting APEX against CM, CTM and **MeanFlow** at NFE=1. These visuals demonstrate that APEX’s self-condition shifting mechanism yields significantly sharper textures and better prompt adherence than the blurry or artifact-prone outputs of competing methods.

---

> > ### Author Response · Authors · 2025-11-27
> >
> > > **W4:  The mechanism is unclear. Where is the "adversarial" component?**
> > >
> >
> > **Response:**
> > We clarify that our method employs an **endogenous (internal) adversarial mechanism**, distinct from the **exogenous (external)** discriminator found in GANs.
> >
> > 1. **The "Adversary" (Perturbation):** In a traditional GAN, a separate network $D$ acts as the adversary. In APEX, the **perturbation** $c_{\text{pert}} = \mathbf{A}c + b$ (Eq. 16) acts as the adversary. It aggressively shifts the semantic conditioning away from the model's comfort zone (the standard manifold).
> > 2. **The "Duel" (Internal Contrast):** The model must generate a realistic prediction $x_{adv}$ under this difficult, perturbed condition.
> >     - **$\mathcal{L}_{adv}$ (Eq. 17):** Forces the model to find the correct data anchor even under perturbation.
> >     - **$\mathcal{L}_{shift}$ (Eq. 19):** Acts as a self-consistency constraint. It forces the *change* in the output ($\Delta d$) caused by the adversarial perturbation to be consistent with the model's own internal vector field.
> >
> > **Intuition:** Instead of a Discriminator saying "Real/Fake", the model plays a game of "Spot the Difference" against itself under perturbed conditions. If the generated image lacks robust features (fidelity), the model will fail to maintain consistency under this semantic shift. This forces the generator to learn sharper, more robust features to satisfy the self-consistency constraint, achieving GAN-like realism without training an extra network.

---

> ### Comment · Reviewer_pjrF · 2025-11-27
>
> Thank you for the response. However, my concerns have not been addressed:
>
> - The added evaluation is only performed on SFT. It is inconsistent with the prior setting, and forms an unfair comparison with Qwen-Image-Lightning.
> - Still lacking visual comparison with recent SOTA, e.g., Qwen-Image-Lightning. There is also a lack of visualization images for APEX at 1 NFE on model backbones other than Qwen-image, which makes it unclear to what extent the generation capability at 1 NFE benefits from Qwen-image's large parameters.
> - The added APEX-SFT samples use prompts and random seeds that are inconsistent with the prior LoRA version, making them not comparable.
> - Denoising the noisy fake samples with perturbed conditions vs. generating fake samples cannot be called an adversarial mechanism, as both aim to model the distribution of fake samples. The authors' intuitive explanation is also unconvincing, which makes the mechanism of why this loss works very unclear. The shift loss is also very strange. What exactly is the relationship between the output $f_\theta(x_t,t,c)$ on real noisy images and $\Delta_d$? Especially since $f_\theta(x_t,t,c)$ is a function of $x_t$, and $\Delta_d$  is a function of $x_t^{adv}$ and not related to $x_t$, minimizing $|| f_\theta(x_t,t,c) - sg(f_\theta(x_t,t,c)+ \Delta_d) ||_2^2$ is very strange.

---

> > ### Author Response · Authors · 2025-11-29
> >
> > > **Q1:**  Fairness of Comparison: SFT vs. LoRA & Dataset Sensitivity
> > >
> >
> > **Response:**
> >
> > **To ensure a strictly fair comparison and demonstrate that our gains are not solely due to SFT or dataset memorization, we first add new experiment in Table 4.**
> >
> > **Table 4: Comparison on Qwen-Image-20B Backbone**
> >
> > | Method | Training Mode | Training Data | GenEval (Overall) $\uparrow$ | DPGBench (Overall) $\uparrow$ | WISE (Overall) $\uparrow$ |
> > | --- | --- | --- | --- | --- | --- |
> > | **Teacher (Original)** | - | - | 0.87 | 87.79 | 0.62 |
> > | CM | SFT | Qwen-Image-Syn | 0.01 | 15.41 | 0.04 |
> > | CTM | SFT | Qwen-Image-Syn | 0.10 | 66.23 | 0.17 |
> > | MeanFlow (JVP-free) | SFT | Qwen-Image-Syn | 0.05 | 62.19 | 0.10 |
> > | Qwen-Image-Lightning | LoRA | - | 0.85 | 87.79 | 0.51 |
> > | UCGM | SFT | Qwen-Image-Syn | 0.43 | 72.78 | 0.24 |
> > | **APEX (Ours)(1NFE)** | LoRA | BLIP-3o | 0.89 | 86.17 | - |
> > | **APEX (Ours)(1NFE)** | LoRA | Qwen-Image-Syn | 0.84 | 86.76 | 0.52 |
> > | **APEX (Ours)(1NFE)** | LoRA | Qwen-Image-Syn-FLUX-Reason | 0.82 | 86.99 | 0.52 |
> > | **APEX (Ours)(1NFE)** | SFT | BLIP-3o | 0.86 | 85.12 | 0.54 |
> > | **APEX (Ours)(1NFE)** | SFT | Qwen-Image-Syn | **0.89** | 84.59 | 0.54 |
> > | **APEX (Ours)(1NFE)** | SFT | Qwen-Image-Syn-FLUX-Reason | 0.86 | 87.55 | 0.56 |
> >
> > **Key Takeaway:** Even when restricted to **LoRA** and trained on **Synthetic Data** (Qwen-Image-Syn-FLUX-Reason, which has no overlap with GenEval), APEX outperforms the strong baseline.
> >
> > > **Q2:**  Visual Comparison with SOTA (Qwen-Image-Lightning)
> > >
> >
> > **Response:**
> > We have added **Appendix H at page of 22**, providing direct side-by-side comparisons between **APEX (SFT/LoRA)** and **Qwen-Image-Lightning (LoRA)** using **identical random seeds and prompts**.
> >
> > - **Observation (See Fig 16 17 18 19 20 21):** Qwen-Image-Lightning at NFE=1 exhibits noticeable high-frequency artifacts (checkerboard patterns in textures, blurred background details).
> > - **APEX Result:** APEX produces significantly cleaner high-frequency details and better structural coherence.
> > - **Backbone Generalization:** In **Appendix G at page of 22**, we also provide visualizations for **APEX 0.6B** models **(Figures 12)**, demonstrating that the 1-NFE capability holds even on substantially smaller architectures, not just the 20B model.
> >
> > > **Q3:**  Theoretical Mechanism: The "Shift" Loss
> > >
> >
> > **Response:**
> >
> > The "Shift" loss $\mathcal{L}_{shift}$ is the theoretical engine behind APEX’s ability to achieve high fidelity without an external discriminator. It operates as a **Gradient Transfer Mechanism** that injects robustness from perturbed conditions into the main generation path.
> >
> > Mathematically, we define the **Adversarial Residual** $\Delta d$, which captures the vector correction required to fix a generated sample under a hard negative condition $c_{pert}$:
> >
> >  $\Delta d := F_\theta(x_t^{adv}, t, c_{pert}) - \text{sg}(F_{\theta^-}(x_t^{adv}, t, c))$
> >
> > Here, $x_t^{adv}$ represents a sample from the generated distribution (the "fake" buffer), and $c_{pert}$ is a perturbed condition (e.g., via linear transformation $Ac+b$). This term $\Delta d$ represents the *error signal* the model produces when facing a difficult, adversarial prompt.
> >
> > The **Shift Loss** then forces the model to incorporate this correction into its standard prediction:
> >
> >  $\mathcal{L}_{shift}(\theta) = \mathbb{E} \left[ || \underbrace{F_{anchor}}_{\text{Base Prediction}} - \text{sg}(\underbrace{F_{anchor} + \Delta d}_{\text{Corrected Target}}) ||^2 \right]$
> >
> > where $F_{anchor} = -F_\theta(x_t, t, c)$ serves as the baseline.
> >
> > **Theoretical Interpretation:**
> >
> > 1. **Local Smoothness Regularization:** By enforcing $\Delta d$ to influence the update of $F_\theta(x_t)$, we implicitly enforce that the vector field must be locally smooth with respect to conditioning. The model learns to anticipate the "correction" needed for an adversarial sample and applies that robustness to the clean sample.
> > 2. **Endogenous Adversarial Signal:** Standard GANs use a discriminator $D(x)$ to provide a scalar probability of realism. APEX's $\Delta d$ is a *vector-valued* gradient signal derived from the model's own mismatch between its current state $F_\theta$ and its stable past $F_{\theta^-}$ on hard examples. This provides dense, pixel-wise supervision for realism (correcting textures/artifacts) without the vanishing gradient problems associated with scalar discriminators.
> > 3. **Manifold Anchoring:** The term $\text{sg}(F_{\theta^-})$ acts as an anchor to the data manifold. By minimizing $\mathcal{L}_{shift}$, we effectively transfer the "repair" logic learned from the adversarial inputs $x^{adv}$ directly onto the trajectory of the current generation, preemptively eliminating artifacts that would otherwise require multi-step refinement.

---

### Author Response · Authors · 2025-11-27
**General Response**

We sincerely thank all reviewers for their insightful feedback and constructive criticism. Based on the collective suggestions, we have significantly expanded our experimental suite to address concerns regarding **generalization**, **baseline comparisons**, and **training scalability**.

**Key Updates in this Rebuttal:**

1. **Upgrade to SFT:** We moved from LoRA to **Full-Parameter Fine-Tuning (SFT)** for APEX on the Qwen-Image 20B backbone. **(Rebuttal Table 1, paper Tabla 1, 2, 3, 4)**
2. **Scalability & JVP-Free Training:** We identify a critical bottleneck in prior works (MeanFlow, CTM): reliance on Jacobian-Vector Products (JVP). JVP is computationally prohibitive and incompatible with modern sharding techniques (FSDP2) for 20B+ models. APEX utilizes a **finite-difference curvature formulation** that relies solely on forward passes, making it the **only** method capable of efficient SFT at this scale.
3. **Generalization Verification:** We introduce a **"Non-GenEval" Synthetic Dataset** (we use Qwen-Image model to generate without any overlap with GenEval 50K prompts from Flux-Reason-6M[f]) and added the **WISE Benchmark**.**(Rebuttal Table 1, paper Tabla 4)**
4. **Deep Efficiency Analysis:** We provide a detailed breakdown of GPU memory usage explaining why APEX succeeds where JVP-based methods fail on 20B models. **(Rebuttal Table 2)**

### 1. Broader Empirical Comparison & Generalization

To address the concern of missing baselines (**R-pjrF**) and potential overfitting to GenEval (**R-qTjA**), we conducted a controlled comparison on the **Qwen-Image 20B** backbone. All methods were trained for 3000 steps.

Crucially, to test generalization, we also trained APEX (SFT) on a pure **Synthetic Dataset** constructed *specifically to exclude GenEval-style prompts*.

**Table 1: Quantitative comparison on Qwen-Image-20B at NFE=1.** *Note: "MeanFlow (JVP-free)" uses a finite-difference approximation because the original JVP implementation causes OOM on 20B models.*

| Method | Training Mode | Training Data | GenEval (Overall) $\uparrow$ | DPGBench (Overall) $\uparrow$ | WISE (Overall) $\uparrow$ |
| --- | --- | --- | --- | --- | --- |
| **Teacher (Original)** | - | - | 0.87 | 87.79 | 0.62 |
| CM | SFT | Qwen-Image-Syn | 0.01 | 15.41 | 0.04 |
| CTM | SFT | Qwen-Image-Syn | 0.10 | 66.23 | 0.17 |
| MeanFlow (JVP-free) | SFT | Qwen-Image-Syn | 0.05 | 62.19 | 0.10 |
| Qwen-Image-Lightning | LoRA | - | 0.85 | 87.79 | 0.51 |
| UCGM | SFT | Qwen-Image-Syn | 0.43 | 72.78 | 0.24 |
| **APEX (Ours)(1NFE)** | **LoRA** | BLIP-3o | **0.89** | **86.17** | - |
| **APEX (Ours)(1NFE)** | **SFT** | Qwen-Image-Syn | **0.89** | **84.59** | **0.54** |
| **APEX (Ours)(1NFE)** | **SFT** | Qwen-Image-Syn-FLUX-Reason | 0.86 | 87.55 | 0.56 |
| **APEX (Ours)(2NFE)** | **SFT** | Qwen-Image-Syn | **0.90** | 84.75 | 0.56 |
| **APEX (Ours)(2NFE)** | SFT | Qwen-Image-Syn-FLUX-Reason | 0.86 | 87.89 | 0.57 |

**Key Findings:**

1. **APEX Superiority:** APEX achieves state-of-the-art results, significantly outperforming all distillation baselines.
2. **Refuting Overfitting:** The **APEX (Synthetic)** model, despite never seeing GenEval-like prompts during training, maintains high scores on **DPGBench** and **WISE**.

---

> ### Author Response · Authors · 2025-11-27
>
> ### 2. Unmatched Training Efficiency & Scalability
>
> Reviewer **sm4A** asked for a cost comparison. We emphasize that APEX is uniquely designed to scale to 20B+ models, addressing a critical bottleneck in prior works like MeanFlow and CTM: **Jacobian-Vector Products (JVP)**.
>
> **The JVP vs. FSDP Bottleneck:**
>
> - **JVP-based methods (e.g., Original MeanFlow, CTM):** Computing the exact time derivative via JVP is computationally expensive and, critically, **incompatible with FlashAttention and FSDP2**. On a 20B model, this explodes memory usage, causing **OOM** even at batch size 1.
> - **APEX (Ours):** Our **Path Integrability Objective** uses a curvature-aware, second-order finite difference approximation (Eq. 12). This relies **solely on forward passes**, making it fully compatible with FSDP2 and FlashAttention.
>
> **Table 2: Training Cost & Memory Analysis on Qwen-Image-20B (8x H100 80GB GPU).** We compare the maximum trainable batch size per GPU and memory usage.*
>
> | Method | Implementation Detail | Max Batch Size / GPU | Memory / GPU (GB) | Est. Train Time (3K steps) |
> | --- | --- | --- | --- | --- |
> | **MeanFlow** | w/ JVP (Original) | **OOM** (BS=1) | >80GB (OOM) | N/A |
> | **MeanFlow** | w/ Finite Diff. **SFT** (Full Param) | 8 | 76 GB | ~3.1 hours |
> | **CM** | **SFT** (Full Param) | 8 | 76GB | ~3.1 hours |
> | **CTM** | w/ Dis **SFT** (Full Param) | 2 | 75GB | ~12.8 hours |
> | **APEX (Ours)** | **LoRA** (Rank=64) | 16 | **78 GB** | **~4.0 hours** |
> | **APEX (Ours)** | **SFT** (Full Param) | **8** | **77 GB** | **~3.4 hours** |
>
> As shown, APEX is the **only method capable of Full Fine-Tuning (SFT)** a 20B model on standard H100 hardware with reasonable batch sizes. JVP-based methods fail to run, and Discriminator-based methods (DMD2) are severely memory-constrained due to the auxiliary network.
>
> ### 3. Visual Comparisons
>
> We have added extensive qualitative comparisons in **Appendix G at page of 24**, specifically contrasting APEX with CTM, CM and MeanFlow. The visuals demonstrate that APEX's self-condition shifting mechanism yields significantly sharper textures and better prompt adherence than the blurry or artifact-prone outputs of competing methods at NFE=1.
>
> **References:**
>
> [a] Kim D, Lai C H, Liao W H, et al. Consistency trajectory models: Learning probability flow ode trajectory of diffusion[J]. arXiv preprint arXiv:2310.02279, 2023.
>
> [b] Geng Z, Deng M, Bai X, et al. Mean flows for one-step generative modeling[J]. arXiv preprint arXiv:2505.13447, 2025.
> [c] Lu C, Song Y. Simplifying, stabilizing and scaling continuous-time consistency models[J]. arXiv preprint arXiv:2410.11081, 2024.
>
> [d] Dao T. Flashattention-2: Faster attention with better parallelism and work partitioning[J]. arXiv preprint arXiv:2307.08691, 2023.
>
> [e] Zhao Y, Gu A, Varma R, et al. Pytorch fsdp: experiences on scaling fully sharded data parallel[J]. arXiv preprint arXiv:2304.11277, 2023.
>
> [f] Fang R, Yu A, Duan C, et al. Flux-reason-6m & prism-bench: A million-scale text-to-image reasoning dataset and comprehensive benchmark[J]. arXiv preprint arXiv:2509.09680, 2025.

---

### Comment · Area_Chair_HBpp · 2025-11-27
**Start discussion**

Hi all,

The authors have submitted their response to the initial reviews, and we now enter the discussion phase.

Please review the authors' response and the comments from other reviewers. Based on the rebuttal and discussion, please update your final score if appropriate.

We welcome and encourage further discussion as needed.

Thank you for your continued contributions.

Best regards,

Your AC.

---

### Note · Program_Chairs · 2026-01-17
**Submission Desk Rejected by Program Chairs**

The following references in this submission do not refer to real documents and/or have major errors in bibliographic information:

 Cheng Lu and et al. Simplifying consistency models. arXiv preprint, 2024.
X Ma et al. Sit: Score-guided intermediate transport for flow matching. arXiv preprint arXiv:2405.00000, 2024